# Adaptive Norm Selection Prevents Catastrophic Overfitting in Fast Adversarial Training

**Fares B. Mehouachi**
New York University in Abu Dhabi
Abu Dhabi, UAE
fm2620@nyu.edu

**Saif Eddin Jabari**
New York University
Abu Dhabi, UAE & Brooklyn, USA
sej7@nyu.edu

## Abstract

We present a novel solution to Catastrophic Overfitting (CO) in fast adversarial training based solely on adaptive $l^p$ norm selection. Unlike existing methods requiring noise injection, regularization, or gradient clipping, our approach dynamically adjusts training norms based on gradient concentration, preventing the vulnerability to multi-step attacks that plagues single-step methods.

We begin with the empirical observation that, with small perturbations, CO occurs predominantly under $l^\infty$ rather than $l^2$ norms. Building on this observation, we formulate generalized $l^p$ attacks as a fixed-point problem and develop $l^p$-FGSM to analyze the $l^2$-to-$l^\infty$ transition. Our key discovery: CO arises when concentrated gradients—with information localized in few dimensions—meet aggressive norm constraints.

We quantify gradient concentration via Participation Ratio from quantum mechanics and entropy metrics, yielding an adaptive $l^p$-FGSM that dynamically adjusts the training norm based on gradient structure. Experiments show our method achieves robust performance without auxiliary regularization or noise injection, offering a principled solution to the CO problem.

## 1 Introduction

Deep neural networks have achieved remarkable success across computer vision, NLP, and speech recognition [1, 2, 3], yet remain vulnerable to adversarial perturbations—subtle input modifications that cause misclassifications [4, 5]. This vulnerability poses important challenges in safety-critical applications including autonomous vehicles [6], healthcare [7], and financial systems [8].

Among defense strategies, adversarial training—incorporating adversarially perturbed examples during training—has proven most effective [5, 9]. However, multi-step methods like Projected Gradient Descent (PGD) [9] impose significant computational costs that limit their applicability in large-scale settings. Fast single-step methods address this efficiency concern but suffer from Catastrophic Overfitting (CO), where models maintain single-step robustness while failing against multi-step attacks [10].

Several approaches have been developed to address CO. RS-FGSM [10] adds uniform random perturbations within the $\epsilon$-ball before applying FGSM, though effectiveness diminishes with larger perturbation radii. GradAlign [11] enforces local linearity by aligning input gradients at clean and adversarial points through double backpropagation, improving robustness but doubling computational overhead. ZeroGrad [12] zeros out small gradient components below a dynamic threshold, preventing overfitting to low-magnitude noise directions with minimal extra cost. N-FGSM [13] removes gradient clipping and uses stronger noise, achieving 3× speedup over GradAlign while maintaining comparable robustness.

39th Conference on Neural Information Processing Systems (NeurIPS 2025).

Recent work has explored CO from various perspectives. AAER [14] identifies "abnormal adversarial examples" where loss decreases during inner maximization and regularizes their occurrence. LAP [15] reveals that pseudo-robust shortcuts form in early network layers, applying adaptive weight perturbations that decrease from former to latter layers. SKG-FAT [16] addresses class imbalance through differentiated class weights and self-knowledge guided label relaxation, achieving 5× speedup over PGD-10. ELLE [17] approximates local linearity regularization without expensive double backpropagation, adapting regularization strength during training. FGSM-PCO [18] prevents inner optimization collapse by generating adversarial examples through adaptive fusion of current and historical perturbations.

While these methods have made important contributions, they typically require auxiliary techniques such as noise injection, regularization, double backpropagation, or architectural modifications. This observation motivates our investigation into whether CO can be addressed through more direct mechanisms.

Our work begins with an empirical observation: CO exhibits interesting norm-dependent behavior. For comparable perturbation amplitudes, $l^\infty$-norm training shows pronounced CO while $l^2$-defense remains more stable, though with limited cross-norm robustness (Figure 1). This suggests that the choice of norm constraint may play a more fundamental role in CO than previously recognized.

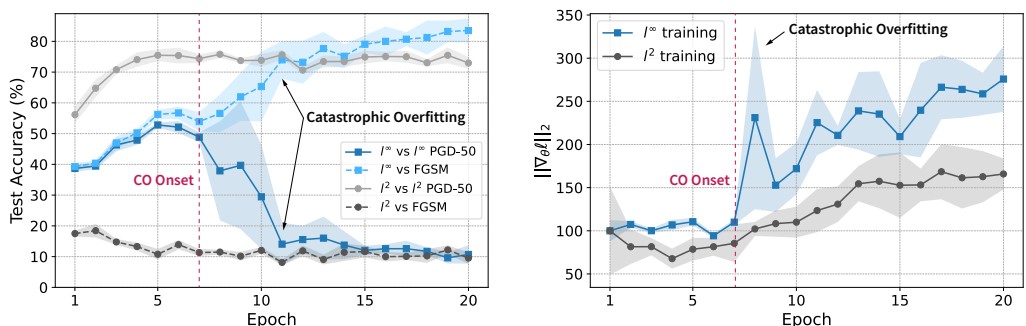

Figure 1: CO phenomena on CIFAR-10 [19] using WideResNet-28-10 [20]: **Left:** $l^\infty$ training ($\epsilon = 8/255$) shows accuracy collapse against PGD-50 [9], while $l^2$ ($\epsilon = 32/255$) remains stable. Legend shows training norm vs attack norm. **Right:** CO onset correlates with gradient norm increase in $l^\infty$ training only.

Building on this observation, we move beyond traditional linear approximations underlying FGSM and adopt a local convexity hypothesis. This leads us to reformulate adversarial attack generation as a fixed-point problem, naturally yielding the $l^p$-FGSM family of attacks. Initial exploration reveals that higher $p$ values ($p \geq 32$) delay but do not prevent CO, while lower values avoid CO at the cost of reduced robustness (Figure 2).

To understand this trade-off, we investigate gradient concentration as a potential mechanism underlying CO. We quantify this through the Participation Ratio (PR) [21, 22]—a measure from quantum mechanics that we adapt to adversarial training as $\text{PR}_1$. Much like its predecessor PR, the adapted metric $\text{PR}_1$ captures how many dimensions meaningfully contribute to gradient magnitude and most importantly connect naturally to the angular separation between $l^2$ and $l^\infty$ bounded perturbations.

Our key insight is that CO emerges when concentrated gradients—with information localized in few dimensions—meet aggressive norm constraints. This concentration can be quantified through participation ratio metrics (detailed in Appendix M), allowing us to adaptively select norm constraints that prevent CO without sacrificing robustness. Based on this understanding, we develop adaptive $l^p$-FGSM that dynamically adjusts the training norm $p$ based on gradient structure. When gradients concentrate (low PR), the method reduces $p$ to maintain better alignment with natural $l^2$ geometry; when gradients distribute more uniformly, higher $p$ values can enhance robustness.

This approach achieves competitive performance on standard benchmarks without requiring noise injection, regularization, or architectural changes. Unlike previous approaches that focus on loss landscapes or gradient alignment, our method directly addresses the gradient concentration phenomenon that precipitates catastrophic overfitting. By providing this connection between gradient geometry and CO, our work offers a complementary perspective suggesting that careful norm selection alone can serve as an effective tool for improving single-step adversarial training.

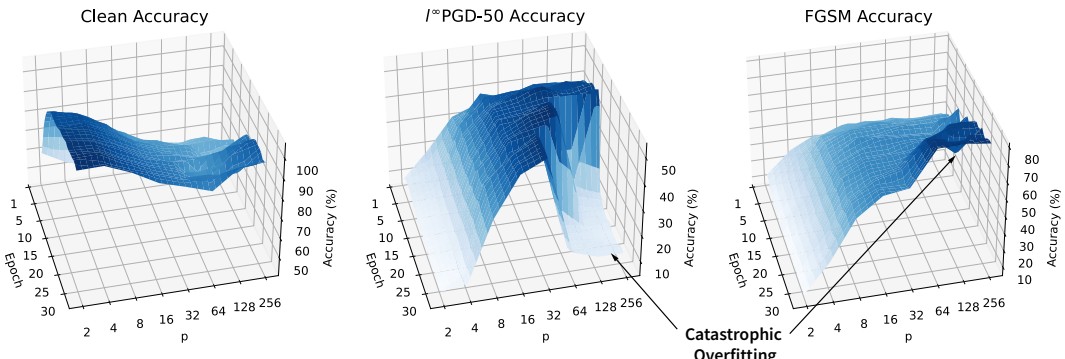

Figure 2: Impact of $l^p$ norm choice on training dynamics and robustness for CIFAR-10 with WideResNet-28-10. The choice of $p$ reveals a key trade-off: higher values ($p \geq 32$) initially show better robustness but become vulnerable to Catastrophic Overfitting (CO), evident in the $l^\infty$ PGD-50 plot (second left). Lower $p$ values prevent CO but with reduced adversarial robustness. Results shown for $\epsilon = 8/255$ over 30 epochs.

## 2   Preliminaries

We consider a classification function $c(x; \theta) : x \mapsto \mathbb{R}^C$ that maps input features $x$ to output logits for classes in set $C$. The prediction probability $\pi_i(x; \theta)$ for label $i$ is given by the softmax function: $\pi_i(x; \theta) = \exp(c_i(x; \theta)) / \sum_j \exp(c_j(x; \theta))$, where $c_i(x; \theta)$ denotes the $i$-th logit and $\theta$ represents model parameters [23].

Adversarial robustness requires that the predicted class remains unchanged under bounded perturbations. Function $c$ is robust to adversarial perturbations of magnitude $\epsilon$ at input $x$ if the class with maximum probability for $x$ retains the highest probability for $x + \delta$, where $\delta$ is any perturbation within the $l^p$ ball of radius $\epsilon$ [4, 5]:

$$\operatorname*{argmax}_{i \in C} \pi_i(x + \delta; \theta) = \operatorname*{argmax}_{i \in C} \pi_i(x; \theta), \ \forall \delta \in B_p(\epsilon) \tag{1}$$

This work considers general $l^p$ norms with $p \geq 2$, using $B(\epsilon)$ to denote $B_p(\epsilon)$ for simplicity.

Standard training employs Empirical Risk Minimization (ERM) [24] over dataset distribution $\mathcal{D}$:

$$\min_\theta \mathbb{E}_{(x,y) \sim \mathcal{D}}[\ell(x; y, \theta)] \tag{2}$$

where $\ell$ represents the loss function, typically cross-entropy $\ell(x; y, \theta) = -y^T \log(\pi(x; \theta))$, and $y$ is the one-hot encoded label. While ERM achieves satisfactory performance on clean data, networks remain vulnerable to adversarial attacks [4, 5], with test accuracy dropping substantially under distributional shifts caused by adversarial perturbations.

Adversarial training [5, 9] addresses this vulnerability by incorporating adversarial examples during training, simulating potential distributional shifts to learn features robust to input perturbations:

$$\min_\theta \mathbb{E}_{(x,y) \sim \mathcal{D}} \left[ \max_{\delta \in B(\epsilon)} \ell(x + \delta; y, \theta) \right] \tag{3}$$

The inner maximization $\max_{\delta \in B(\epsilon)} \ell(x + \delta; y, \theta)$ is typically approximated through gradient-based optimization. Projected Gradient Descent (PGD) [9] performs iterative updates:

$$\delta \leftarrow \Pi \left( \delta - \mu \nabla_x \ell(x + \delta; y, \theta) \right) \tag{4}$$

where projection operator $\Pi$ ensures perturbations remain within bounds through scaling ($l^2$) or clipping ($l^\infty$).

Multi-step methods like PGD incur significant computational costs. The Fast Gradient Sign Method (FGSM) [5] provides efficiency through first-order Taylor expansion $\ell(x_0 + \delta) \approx \ell(x_0) + \delta^T \nabla_x \ell$,

using gradient sign to solve the maximization problem:

$$\delta_{\text{FGSM}} = \operatorname*{argmax}_{\delta \in B_\infty(\epsilon)} \left( \ell(x_0) + \delta^T \nabla_x \ell \right) = \epsilon \operatorname{sign}(\nabla_x \ell) \tag{5}$$

While FGSM efficiently solves the linearized maximization problem in Eq. (3) under $l^\infty$ constraints, it suffers from Catastrophic Overfitting (CO). Wong et al. [10] proposed adding random noise $\eta \sim \mathcal{U}[-\epsilon, \epsilon]$ as remedy:

$$\delta_{\text{RS-FGSM}} = \Pi_{B_\infty(\epsilon)} \left( \eta + \epsilon \operatorname{sign}(\nabla_x \ell(x_0 + \eta)) \right) \tag{6}$$

Our work extends beyond first-order approximations by characterizing the inner maximization in Eq. (3) under general $l^p$ constraints, leading to a fixed-point formulation.

## 3 Theoretical Framework

We develop a theoretical foundation that moves beyond the local linearity assumption underlying FGSM by adopting a local convexity framework. This perspective reveals that optimal perturbations reside on constraint boundaries, enabling our fixed-point formulation for general $l^p$ norms and providing the mathematical foundation for preventing catastrophic overfitting through principled norm selection.[1] [2]

Under local convexity, optimal adversarial perturbations are guaranteed to lie on the boundary $\partial B_p(\epsilon)$, as any interior critical point must be a local minimum when the Hessian $\nabla_x^2 \ell$ is positive definite. We demonstrate that this condition emerges naturally during training through Hessian analysis and empirical validation (detailed in Appendix A). This enables controlled transitions between the catastrophic overfitting-resistant $l^2$ regime and the catastrophic overfitting-prone $l^\infty$ regime.

### 3.1 $l^2$ Norm-Bounded Adversarial Attacks

Given that optimal perturbations exist on the boundary under local convexity, we use Lagrange multipliers to reformulate the constrained maximization problem in Eq. (3) as an unconstrained optimization, leading to a fixed-point characterization.

**Proposition 1.** *For a training sample $x_0$ with non-null gradient, the optimal perturbation $\delta^\star$ within $B_2(\epsilon)$ exists and solves the fixed-point problem $\delta^\star = F(\delta^\star)$, where:*

$$F(\delta) = \epsilon \frac{\nabla_x \ell(x_0 + \delta)}{\|\nabla_x \ell(x_0 + \delta)\|_2} \tag{7}$$

*$F$ is Lipschitz continuous around its origin with constant $K = 2\epsilon \|\nabla_x^2 \ell\| / \|\nabla_x \ell(x_0)\|_2$:*

$$\|F(\delta) - F(0)\| \leq K \|\delta\| \tag{8}$$

*and the fixed-point problem converges if $K < 1$.*

**Proof.** See Appendix B. □

Equation (7) defines a fixed-point iteration that approximates the optimal perturbation, as illustrated in Figure 3. The Lipschitz constant $K$ connects to curvature control techniques: CURE [25] minimizes Hessian norms for robustness, while Srinivas et al. [26] introduced gradient norm division for scale-invariant curvature. Reducing $K$ accelerates convergence of the inner maximization in Eq. (3).

**Corollary (GradAlign Connection).** *When $\nabla_x \ell(x_0)$ aligns with $\nabla_x \ell(x_0 + \epsilon \nabla_x \ell / \|\nabla_x \ell\|)$, the fixed-point converges instantly[3]:*

$$\frac{\nabla_x \ell(x_0 + \epsilon \nabla_x \ell / \|\nabla_x \ell\|)}{\|\nabla_x \ell(x_0 + \epsilon \nabla_x \ell / \|\nabla_x \ell\|)\|} = \frac{\nabla_x \ell}{\|\nabla_x \ell\|} \tag{9}$$

GradAlign [11] regularizes gradient alignment, effectively improving the initialization of our fixed-point algorithm, explaining its empirical success.

---

[1] If local convexity does not hold, the framework gracefully defaults to the standard local linearity approach.

[2] For one-step adversarial training, local linearity and convexity lead to identical outcomes.

[3] In this ideal case, the normalized gradient is already the fixed point.

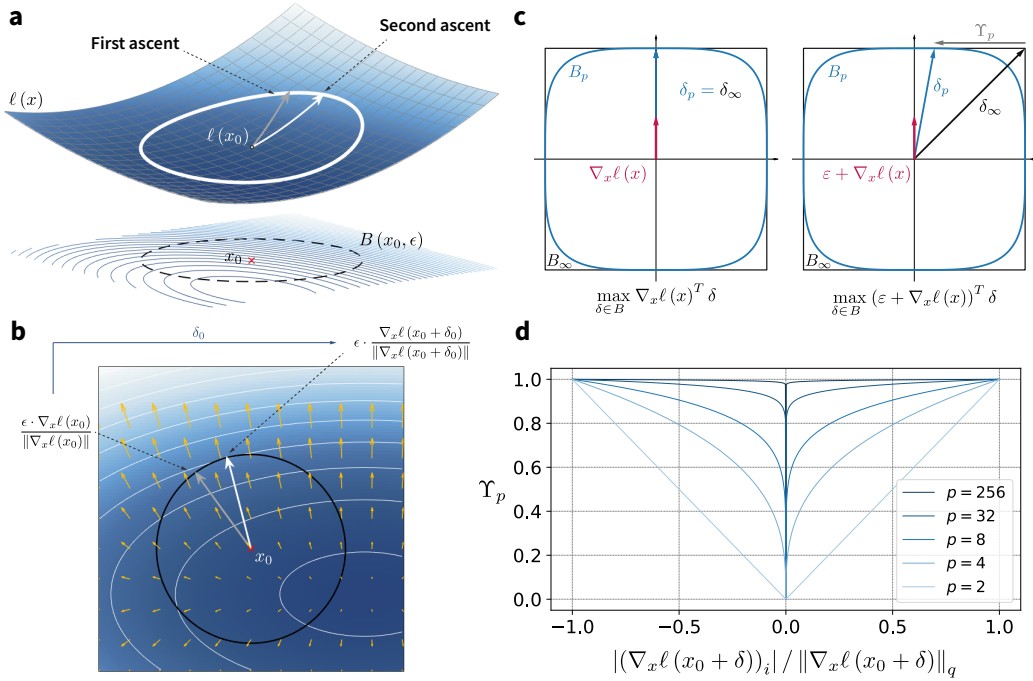

Figure 3: Geometric interpretation of $l^p$-FGSM framework. **(a,b)** Fixed-point algorithm iterations for optimal perturbation identification under $l^2$ constraint (Eq. 7). **(c)** Attack geometry under different $l^p$ norms: *Left* - ideal scenario with aligned gradients; *Right* - effect of gradient noise showing $l^\infty$ sensitivity versus $l^p$ stability. **(d)** Transition function $\Upsilon_p$ variation across $p$ values, demonstrating smooth high-pass filtering behavior.

## 3.2 $l^p$ Norm-Bounded Adversarial Attacks

We extend the fixed-point framework to general $l^p$ norms, which serve as smooth interpolations between $l^2$ and $l^\infty$. This extension enables our approach to catastrophic overfitting through controlled norm transitions based on gradient structure.

**Proposition 2.** *For a training sample $x_0$ with non-null gradient under $B_p(\epsilon)$ constraint, the optimal perturbation $\delta^\star$ exists and solves the fixed-point equation $\delta^\star = F_p(\delta^\star)$, where:*

$$F_p(\delta) = \epsilon\, sign(\nabla_x\ell(x_0 + \delta)) \left| \frac{\nabla_x\ell(x_0 + \delta)}{\|\nabla_x\ell(x_0 + \delta)\|_q} \right|^{q-1} \tag{10}$$

*with $l^q$ being the dual norm of $l^p$: $\frac{1}{p} + \frac{1}{q} = 1$. All operations are element-wise.*

**Proof.** See Appendix C. □

**Unified Attack Spectrum:** Equation (10) provides a unified formulation spanning from $l^2$ to $l^\infty$. For $p = q = 2$, we recover Eq. (7); as $p \to \infty$, we obtain $q = 1$ and recover FGSM. The transition between regimes is governed by:

$$\Upsilon_p(\delta) = \left| \frac{\nabla_x\ell(x_0 + \delta)}{\|\nabla_x\ell(x_0 + \delta)\|_q} \right|^{q-1} \tag{11}$$

This function acts as a smooth high-pass filter, approaching unity everywhere except near zero (Figure 3d). Unlike discontinuous thresholding in ZeroGrad [12], our approach provides smooth gradient filtering that preserves differentiability and training stability.

**Convergence Analysis:** For $p > 2$, global Lipschitz continuity fails due to the discontinuous sign function and concave power term $(q-1)$ near zero gradients. However, we ensure local Lipschitzness by maintaining gradients bounded away from zero:

$$\exists m > 0 : \forall i, \forall \delta \in \partial B_p(\epsilon), |\nabla_x\ell(x_0 + \delta)_i| > m \tag{12}$$

This condition motivates our algorithmic design: adding constant $\varepsilon$ to gradient components ensures both numerical stability and theoretical convergence guarantees. Under this modification, $F_p$ becomes locally Lipschitz with constant $K(p, m)$ (detailed in Appendix D).

### 3.3 Gradient-Aware Adaptive Norm Selection

While fixed $p$ values can balance robustness and stability, our preliminary analysis reveals fundamental limitations. As detailed in Appendix E, higher $p$ values delay catastrophic overfitting but eventually succumb to it, while lower $p$ values provide stability at the cost of reduced robustness. This fundamental trade-off varies significantly across datasets, with dataset complexity critically influencing optimal $p$ selection, motivating our adaptive approach.

**High-Dimensional Perturbation Analysis:** The choice of norm becomes increasingly critical as dimensionality grows. In $\mathbb{R}^d$, perturbation amplitudes scale directly with dimension:[4]

$$\|\delta_2\|_2 = \epsilon, \ \|\delta_\infty\|_2 \overset{a.s.}{=} \epsilon\, d^{1/2}, \ \max\|\delta_p\|_2 = \epsilon\, d^{(1/2-1/p)} \tag{13}$$

These relationships, which appear in adversarial PAC-Bayes bounds [27], reveal that $l^\infty$-bounded perturbations yield vectors dramatically distant from original samples as dimension increases. For CIFAR-10 ($d = 3{,}072$) and ImageNet ($d \sim 1.5 \times 10^5$), this effect becomes particularly significant.

Our key insight: reducing $p$ effectively constrains the perturbation space from dimension $d$ to an effective dimension $d_e$, where $d^{(1/2-1/p)} \sim d_e^{1/2}$. This suggests that measuring the intrinsic effective dimension of gradients can guide appropriate $p$ selection.

**Participation Ratio for Gradient Concentration:** We adapt the Participation Ratio from quantum mechanics [21, 22], which quantifies electron localization, to measure gradient concentration:

$$\mathrm{PR}(x) = \frac{(\sum_i |x_i|^2)^2}{\sum_i |x_i|^4} = \left(\frac{\|x\|_2}{\|x\|_4}\right)^4 \tag{14}$$

For adversarial training, we substitute the standard ones vector with the gradient's sign vector, yielding:

$$\mathrm{PR}_1 = \left(\frac{\|\nabla_x\ell\|_1}{\|\nabla_x\ell\|_2}\right)^2 \tag{15}$$

This effective dimension varies between 1 and $d$ for non-null vectors and naturally connects to the angular separation between $\delta_2$ and $\delta_\infty$ attacks:

$$\cos(\theta_{2,\infty}) = \frac{\|\nabla_x\ell\|_1}{\|\nabla_x\ell\|_2 d^{1/2}} = \sqrt{\frac{\mathrm{PR}_1}{d}} \tag{16}$$

Figure 4 provides empirical validation of our theoretical framework. Both participation ratios drop sharply at CO onset, with corresponding increases in angular separation between $l^2$ and $l^\infty$ perturbations. This confirms gradient concentration's role in triggering catastrophic behavior and validates our adaptive norm selection strategy.

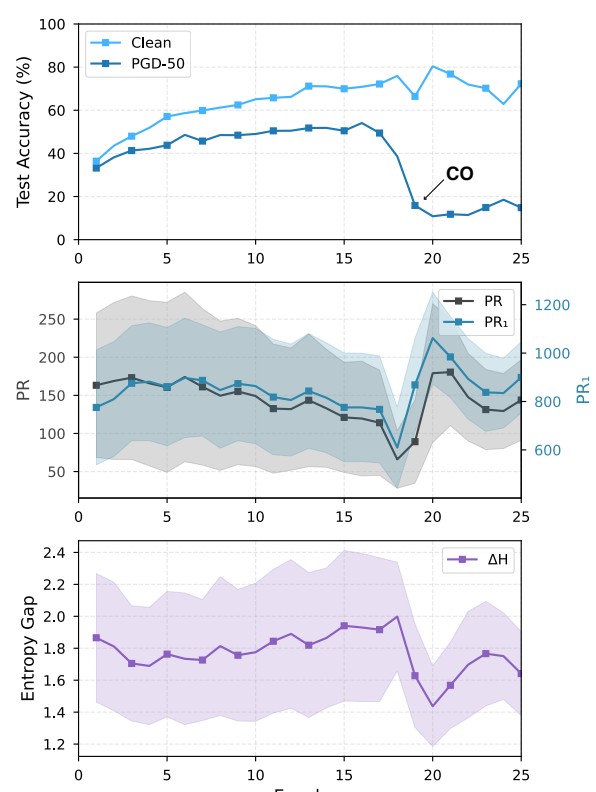

Figure 4: Evolution of Participation Ratios ($\mathrm{PR}$, $\mathrm{PR}_1$) and entropy gap during training. Sharp declines in these metrics precisely align with Catastrophic Overfitting (CO) onset, demonstrating how gradient concentration directly precedes and triggers adversarial vulnerability.

---

[4]For $p > 2$, maximum occurs when all components have equal amplitude.

**Noise-Induced Alignment:** Classical CO remedies involve noise injection [10, 13]. Our framework shows that noise increases $PR_1$, enhancing alignment between $l^\infty$ and $l^2$ attacks:

**Lemma 1** (Noise-Induced Alignment). *For normalized gradient $g = \nabla_x \ell / \|\nabla_x \ell\|_2$ and additive zero-mean noise $\eta \sim \mathcal{U}[-M, M]^d$, there exists $\alpha > 0$ such that if $M < \alpha \|g\|_\infty$, then:*

$$\mathbb{E}\left[\frac{\|g + \eta\|_1}{\|g + \eta\|_2}\right] \geq \frac{\|g\|_1}{\|g\|_2} \tag{17}$$

**Proof.** See Appendix F. □

**Monotonic Angular Relationships:** We establish that norm reduction systematically improves angular alignment:

**Lemma 2** (Monotonicity of Angular Separation). *For any non-null gradient $\nabla_x \ell$ and $p \geq 2$, the cosine between $l^2$ and $l^p$ perturbations satisfies:*

$$\cos(\theta_{2,\infty}) \leq \cos(\theta_{2,p}) \quad where \quad \cos(\theta_{2,p}) = \frac{\|\nabla_x \ell\|_q^q}{\|\nabla_x \ell\|_2 \|\nabla_x \ell\|_{2(q-1)}^{q-1}} \tag{18}$$

**Proof.** See Appendix G. □

**Entropy-Based Norm Selection:** Direct computation of optimal $p$ from Eq. (18) proves challenging. For $q \in [1, 2]$ and moderate increases, first-order Taylor expansion provides computational efficiency (details in Appendix H):

$$\cos(\theta_{2,p}) = \sqrt{\frac{PR_1}{d}}\left(1 + (q-1)\Delta H\right) + \mathcal{O}((q-1)^2) \tag{19}$$

where $\Delta H = H_m - H$ is the entropy gap between logarithmic mean entropy $H_m$ and Shannon entropy $H$ of normalized gradient components:

$$H = -\sum_{i=1}^{d} \rho_i \log(\rho_i), \quad H_m = -\log \prod_{i=1}^{d} (\rho_i)^{1/d}, \quad \rho_i = \frac{|\nabla_x \ell_i|}{\|\nabla_x \ell\|_1} \tag{20}$$

Setting a threshold $\tau$ below which cosine alignment should not drop, we derive:

$$q^* \geq 1 + \frac{(\tau\sqrt{d/PR_1} - 1)}{\Delta H}, \; \tau \in [0, 1] \tag{21}$$

This formula captures the interplay between gradient geometry and norm selection: when gradients concentrate (low $PR_1$) and entropy gap decreases, $q$ increases (lower $p$) to maintain alignment. For practical implementation:

$$\tau \equiv (1 + \alpha)\cos(\theta_{2,\infty}) \equiv \cos((1 - \beta)\theta_{2,\infty}) \tag{22}$$

These theoretical insights directly inform our algorithmic design. By dynamically adjusting $p$ based on gradient concentration metrics $PR_1$ and entropy gap, we maintain alignment with natural $l^2$ geometry when gradients concentrate (low $PR_1$) and increase $p$ when gradients distribute uniformly (high $PR_1$). This adaptive approach prevents concentrated gradients from meeting aggressive norm constraints—precisely the condition triggering CO.

### 3.4 $l^p$-FGSM Algorithm

Our $l^p$-FGSM algorithm performs one fixed-point iteration ($\delta^{(1)} = F_p(\delta^{(0)})$) with zero initialization, maintaining computational efficiency while accessing the full spectrum of $l^p$ attack geometries. The epsilon stabilization step serves dual purposes: ensuring numerical stability and satisfying the Lipschitz conditions in Eq. (12).

The adaptive norm selection mechanism automatically adjusts $p$ based on gradient concentration statistics, enabling transitions between attack geometries as training progresses. When gradients concentrate (indicating potential CO onset), the algorithm reduces $p$ to maintain alignment with natural $l^2$ geometry. When gradients distribute uniformly, higher $p$ values enhance robustness.

This theoretical framework establishes that adaptive norm selection is mathematically sound, maintains convergence properties, and provides a principled solution to catastrophic overfitting without auxiliary techniques like noise injection or regularization.

---

**Algorithm 1** $l^p$-FGSM

---

1: **Input:** Model $\theta$, data $x$, labels $y$, loss $\ell$, optimizer, attack amplitude $\epsilon$, norm $p$ (dual $q$)
2: **repeat**
3:     Sample minibatch $(x_0, y_0)$
4:     Compute gradient $g_x \leftarrow \nabla_{x_0} \ell(x_0, y_0)$
5:     Apply stability term: $\bar{g}_x \leftarrow \varepsilon + |g_x|$
6:     **if** adaptive **then** Update $q$ via Eq. 21 using $\mathrm{PR}_1$, $\Delta H$
7:     Compute attack $\delta_p \leftarrow \epsilon \cdot \mathrm{sign}(g_x) \cdot |\bar{g}_x / \|\bar{g}_x\|_q|^{q-1}$
8:     Update $\theta$ with $\nabla_\theta \ell(x_0 + \delta_p, y_0)$ and optimizer
9: **until** Convergence criteria
10: **Output:** Robust model $\theta$

---

## 4 Experiments and Results

Our $l^p$-FGSM approach provides computational efficiency over methods requiring double backpropagation, with overhead limited to gradient norm calculations. We evaluate our method on standard datasets, examine norm selection and gradient concentration relationships, and compare against state-of-the-art fast adversarial training methods.

### 4.1 Comparison with Benchmark Techniques

To rigorously evaluate the effectiveness of adaptive $l^p$-FGSM, we conducted comprehensive comparisons against several well-established fast adversarial training methods, including RS-FGSM [10], ZeroGrad [12], N-FGSM [13], and GradAlign [11]. This diverse subset, representing fundamentally different conceptual approaches to addressing CO, provides a robust basis for assessing the capacity of adaptive $l^p$ norms to mitigate the phenomenon while maintaining adversarial robustness. For consistency and fair comparison, we used the recommended hyperparameters for each benchmark method as specified in their respective publications.

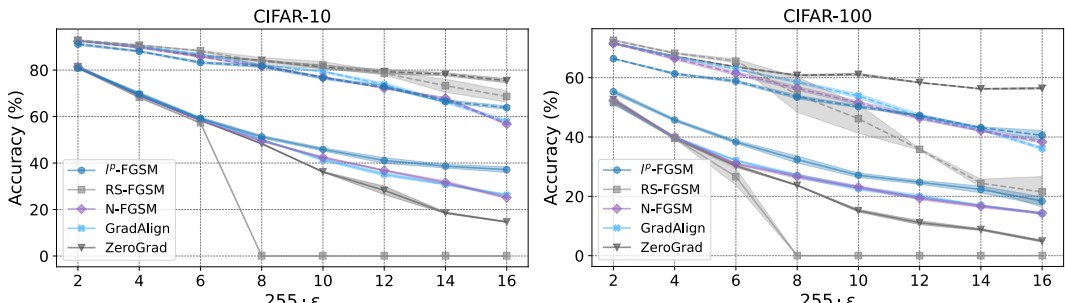

Figure 5: Performance benchmarking of adaptive $l^p$ norm-based training against single-step and fast adversarial techniques using PGD-50-10, demonstrating the competitive efficacy of adaptive $l^p$-FGSM. Results were achieved with an SGD optimizer with a cosine learning rate schedule (30 epochs, minimum 0.001, maximum 0.2), weight decay of $5 \cdot 10^{-4}$, and a dropout rate of 0.1. For CIFAR-10, $\beta = 0.01$ was applied, while for CIFAR-100, $\beta = 0.1$ was used (Eq. 22). We switched from ADAM to SGD for these comparisons as it is the standard optimizer in adversarial training literature and facilitates direct comparison with published results.

Our empirical studies, summarized in Figure 5, demonstrate that adaptive $l^p$-FGSM not only meets but often surpasses the robustness benchmarks of leading fast methods [9, 25, 28, 11, 13]. This success hinges on the choice of the $l^p$ norm, which enhances robustness against $l^\infty$ attacks while resolving CO without requiring noise injection or expensive regularization. All components of $l^p$-FGSM (Alg. 1) are efficient to compute with minimal overhead, making the approach particularly attractive for large-scale applications where computational efficiency is a priority.

The performance advantage of our method is particularly pronounced at higher perturbation magnitudes ($\epsilon \geq 8/255$), where many competing approaches suffer from CO or significant robustness degradation. This innovative use of norm selection introduces a simple yet effective approach to fast adversarial training, offering a novel perspective to advance robust machine learning.

## 4.2 Experiments with ImageNet

To evaluate adaptive $l^p$-FGSM on high-resolution images representative of real-world applications, we conducted extensive experiments on ImageNet-1k [29], training a pre-trained ResNet-50 model with ADAM optimizer (lr=$10^{-4}$, batch size 128) for 15 epochs. We tested our method ($\beta = 0.1$, $\varepsilon = 10^{-12}$) against PGD-50 attacks across a range of perturbation magnitudes $\epsilon = (2, 4, 6)/255$ and compared with established methods including FGSM, RS-FGSM, and N-FGSM.

As shown in Table 1, while FGSM experiences catastrophic overfitting at $\epsilon = 6/255$ (evidenced by the near-zero adversarial accuracy), adaptive $l^p$-FGSM achieves superior adversarial robustness across all perturbation levels while maintaining competitive clean accuracy. The performance advantage is particularly significant at $\epsilon = 4/255$ and $\epsilon = 6/255$, where our method outperforms RS-FGSM by 3.23% and 3.30% in adversarial accuracy, respectively.

Table 1: Comparative Analysis of Robustness Against PGD-50-10 on ImageNet-1k. FGSM, RS-FGSM and N-FGSM results are from [13]. All methods utilize ImageNet-1k pre-trained weights and undergo 15 epochs of training. Results show clean accuracy (top) and PGD-50 accuracy (bottom).

| ImageNet-1k ResNet-50 | | | |
|---|---|---|---|
| **Method** | $\epsilon = 2/255$ | $\epsilon = 4/255$ | $\epsilon = 6/255$ |
| FGSM | 54.72% | 48.50% | 48.55% |
| | **38.21%** | 25.86% | 0.08% |
| RS-FGSM | **56.29%** | **50.81%** | 47.67% |
| | 36.86% | 25.12% | 16.49% |
| $l^p$-FGSM | 53.18% | 48.42% | **48.61%** |
| | 37.94% | **28.35%** | **19.79%** |
| N-FGSM | 54.39% | 47.56% | 47.70% |
| | 38.07% | 26.28% | 17.12% |

These results on ImageNet-1k demonstrate the scalability of our approach to large, complex datasets and its effectiveness in addressing CO in practical settings. The consistent performance advantages across different perturbation magnitudes highlight the robustness of the adaptive norm selection strategy in diverse scenarios, reinforcing the potential of $l^p$-FGSM as a general-purpose solution for fast adversarial training.

## 5 Conclusion

We presented adaptive $l^p$-FGSM, a principled approach to mitigating catastrophic overfitting in fast adversarial training. Our investigation began with the observed discrepancy between $l^2$ and $l^\infty$ norms, motivating us to explore the full $l^p$ spectrum between these extremes. This led to reformulating adversarial attack generation as a fixed-point problem, enabling efficient single-step methods while providing theoretical insights through Lipschitz continuity analysis. While our approach relies on local convexity assumptions, it gracefully defaults to local linearity when these assumptions do not hold.

Our key finding—that catastrophic overfitting emerges when concentrated gradients meet aggressive norm constraints—provides a unifying perspective on previous observations. By adapting the Participation Ratio from quantum mechanics to measure both gradient concentration and angular separation, we established a quantitative connection between gradient geometry and adversarial vulnerability. This insight led to dynamically adjusting the training norm $p$ based on gradient structure. Although our method avoids double backpropagation, it still requires hyperparameters for angle constraints that warrant further optimization across different architectures and datasets. Future work could explore extending our adaptive framework to defend against mixed-norm attacks that combine multiple $l^p$ constraints.

This work contributes to understanding fast adversarial training by connecting gradient geometry to training dynamics through an information-theoretic lens. By establishing adaptive norm selection as a theoretically motivated approach, we hope to inspire further research into geometric perspectives on adversarial robustness. Our results suggest that careful consideration of gradient structure may be essential in developing efficient and robust training methods. By establishing a theoretical foundation for addressing catastrophic overfitting, our work contributes to the broader goal of developing reliable machine learning systems that maintain robustness guarantees even under computational constraints.

## Code Availability

The code for $l^p$-FGSM is available at `https://github.com/FaresBMehouachi/lpfgsm`. The authors declare no competing interests.

## Acknowledgment

This work was supported by the NYUAD Center for Interacting Urban Networks (CITIES), funded by Tamkeen under the NYUAD Research Institute Award CG001. The views expressed in this article are those of the authors and do not reflect the opinions of CITIES or their funding agencies

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

# A  Local Convexity Analysis

In this appendix, we provide a detailed analysis of the local convexity framework that underlies our $l^p$-FGSM approach. We examine both the theoretical foundations and empirical evidence for local convexity emergence during adversarial training.

## A.1  Theoretical Foundation of Local Convexity

While fast adversarial training traditionally relies on local linearity assumptions through first-order Taylor expansions, we examine a more general local convexity framework that emerges from analyzing the Hessian of the loss function with respect to inputs. When the Hessian $\nabla_x^2 \ell$ is positive definite, any critical point in the perturbation ball's interior must be a local minimum, forcing the maximum to occur on the boundary $\partial B_p(\epsilon)$—a useful property that enables efficient single-step methods.

The Hessian structure can be decomposed with respect to the output logits as:

$$\nabla_x^2 \ell = \left( \frac{\partial \pi}{\partial x_0} \right) \frac{\partial^2 \ell}{\partial \pi^2} \left( \frac{\partial \pi}{\partial x_0} \right)^T + \frac{\partial^2 \pi}{\partial x_0^2} \frac{\partial \ell}{\partial \pi} \tag{23}$$

This decomposition reveals two distinct components:

**Gauss-Newton Term:** The first term $\left( \frac{\partial \pi}{\partial x_0} \right) \frac{\partial^2 \ell}{\partial \pi^2} \left( \frac{\partial \pi}{\partial x_0} \right)^T$ is positive semi-definite since $\frac{\partial^2 \ell}{\partial \pi^2}$ represents the Hessian of the cross-entropy loss with respect to predictions, which is always positive definite for proper probability distributions.

**Error-Dependent Term:** The second term $\frac{\partial^2 \pi}{\partial x_0^2} \frac{\partial \ell}{\partial \pi}$ involves the prediction errors $\frac{\partial \ell}{\partial \pi}$. As training progresses and the model's predictions improve, these error terms diminish, reducing the magnitude of the second term relative to the first.

## A.2  Convergence to Local Convexity During Training

The natural emergence of local convexity during training can be understood through the evolution of the Hessian structure in Eq. (23). As the model learns to minimize the training loss, the prediction errors $\frac{\partial \ell}{\partial \pi}$ systematically decrease. This causes the potentially indefinite second term to diminish in magnitude relative to the positive semi-definite Gauss-Newton term, leading to an overall positive definite Hessian.

This convergence can be accelerated through architectural choices that control the second-order derivatives $\frac{\partial^2 \pi}{\partial x_0^2}$:

**Activation Function Selection:** Smooth activation functions like SELU [30] or GELU [31] have well-behaved second derivatives, leading to more stable convergence to local convexity compared to non-smooth activations.

**Network Depth and Width:** Deeper networks tend to develop local convexity more readily as the composition of smooth functions preserves convexity properties under appropriate conditions.

However, our empirical analysis demonstrates that even standard ReLU networks, despite their non-smooth activation functions, naturally develop local convexity through the training process, as visualized in Figure 6.

## A.3  Empirical Evidence for Local Convexity

Figure 6 provides empirical validation of the local convexity emergence during training. The visualization shows the loss landscape around training points at different stages of the training process.

**Early Training (Upper Panels):** After one epoch, the loss landscapes exhibit irregular, non-convex characteristics with multiple local minima and saddle points. The landscapes are complex and do not satisfy the local convexity assumption.

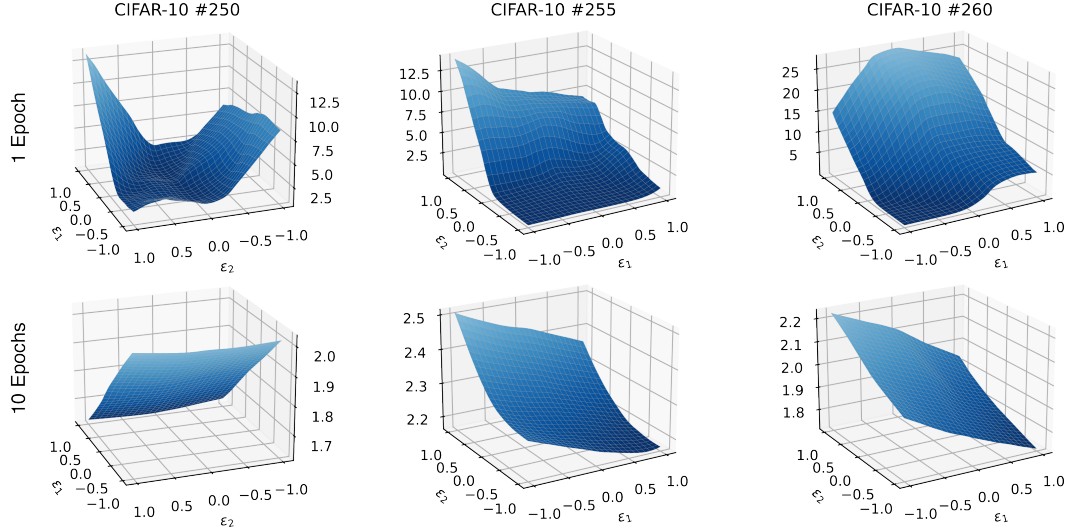

Figure 6: Empirical evidence for local convexity emergence during training on CIFAR-10. The upper panels display the loss landscape after one epoch of training, while the lower panels show the same landscape after ten epochs with $l^p$-FGSM training. Training points are positioned at $(0, 0)$; $\varepsilon_1$ and $\varepsilon_2$ are eigenvectors corresponding to the extreme eigenvalues of the input Hessian $\nabla_x^2 \ell$ for each sample. The progressive development of convex loss landscapes validates our theoretical framework and provides justification for boundary-focused adversarial search strategies.

**Later Training (Lower Panels):** After ten epochs of training, the same landscapes show clear convex structure around the training points. The loss increases monotonically as we move away from the training point in any direction within the neighborhood, confirming positive definiteness of the local Hessian.

This empirical observation has several important implications:

1. **Theoretical Validation:** The emergence of local convexity validates our theoretical framework and justifies the use of boundary-focused optimization strategies.

2. **Practical Robustness:** Even when local convexity does not hold initially, the framework can gracefully default to local linearity assumptions, ensuring robustness across different training phases.

3. **Efficient Optimization:** The development of local convexity enables more efficient single-step adversarial example generation, as the optimal perturbations are guaranteed to be on the constraint boundary.

# B  Appendix: Demonstration $l^2$ Optimal Attack

**Proposition 1.** *Consider a training sample $x_0$ with a non-null gradient. The optimal perturbation $\delta^\star$ within $B(\epsilon)$ exists and corresponds to the solution of a fixed-point problem $\delta^\star = F(\delta^\star)$, where*

$$F(\delta) = \epsilon \frac{\nabla_x \ell(x_0 + \delta)}{\|\nabla_x \ell(x_0 + \delta)\|_2} \tag{24}$$

*The function $F$ exhibits Lipschitzian behavior around its origin, satisfying:*

$$\|F(\delta) - F(0)\| \leq 2\epsilon \frac{\|\nabla_x^2 \ell\|}{\|\nabla_x \ell(x_0)\|_2} \|\delta\| \tag{25}$$

*The fixed-point problem converges if it is contractive:*

$$K = 2\epsilon \frac{\|\nabla_x^2 \ell\|}{\|\nabla_x \ell(x_0)\|_2} < 1 \tag{26}$$

*Proof.* Assuming that the Hessian of the loss function, $\nabla_x^2 \ell$, is positive definite, any critical point in the interior would be a minimum. The implicitly assumed compactness guarantees the existence of the maximum on the boundary. The constrained maximization uses the Lagrangian:

$$\mathcal{L}(\delta, \lambda) = \ell(x_0 + \delta) - \frac{\lambda}{2}(\delta^T \delta - \epsilon^2) \tag{27}$$

The derivatives yield the following equations:

$$\left\{ \frac{\partial}{\partial \delta} \mathcal{L} = \nabla_x \ell(x_0 + \delta) - \lambda \delta = 0 \, \frac{\partial}{\partial \lambda} \mathcal{L} = -\frac{1}{2}(\delta^T \delta - \epsilon^2) = 0 \right. \tag{28}$$

Since the maximum exists on the boundary, the constraint $\delta^T \delta = \epsilon^2$ is activated; hence the Lagrange multiplier $\lambda$ is non-null. The gradient at $x_0 + \delta$ cannot be null (minimum otherwise), therefore $\|\nabla_x \ell(x_0 + \delta)\| > 0$.

Solving the two Lagrangian equations yields:

$$\delta = \pm \epsilon \frac{\nabla_x \ell(x_0 + \delta)}{\|\nabla_x \ell(x_0 + \delta)\|} \tag{29}$$

Given the positive Hessian assumption, moving along the gradient (equivalent to choosing the positive sign) results in a greater change in the loss function $\ell$. Consequently:

$$\delta = \epsilon \frac{\nabla_x \ell(x_0 + \delta)}{\|\nabla_x \ell(x_0 + \delta)\|} \tag{30}$$

The maximum $\delta^\star$ is the solution to a fixed-point problem. The existence and uniqueness of the solution $\delta^\star$ is guaranteed if $F(\delta)$ is contractive, i.e., Lipschitz continuous with a Lipschitz constant $K < 1$.

To demonstrate this Lipschitz continuity, we consider:

$$\|F(\delta) - F(0)\| = \epsilon \left\| \frac{\nabla_x \ell(x_0 + \delta)}{\|\nabla_x \ell(x_0 + \delta)\|} - \frac{\nabla_x \ell(x_0)}{\|\nabla_x \ell(x_0)\|} \right\|$$

By introducing a cross term and using the triangle inequality:

$$\|F(\delta) - F(0)\| \leq \epsilon \left\| \frac{\nabla_x \ell(x_0)}{\|\nabla_x \ell(x_0)\|} - \frac{\nabla_x \ell(x_0 + \delta)}{\|\nabla_x \ell(x_0)\|} \right\| + \epsilon \left\| \frac{\nabla_x \ell(x_0 + \delta)}{\|\nabla_x \ell(x_0)\|} - \frac{\nabla_x \ell(x_0 + \delta)}{\|\nabla_x \ell(x_0 + \delta)\|} \right\|$$

The first term can be bounded:

$$\|F(\delta_1) - F(0)\| \leq \epsilon \frac{\|\nabla_x^2 \ell(x_0)\| \|\delta\|}{\|\nabla_x \ell(x_0)\|} + \epsilon \|\nabla_x \ell(x_0 + \delta)\| \left| \frac{1}{\|\nabla_x \ell(x_0 + \delta)\|} - \frac{1}{\|\nabla_x \ell(x_0)\|} \right|$$

After unifying the denominator:

$$\|F(\delta) - F(0)\| \leq \epsilon \frac{\|\nabla_x^2 \ell\|\|\delta\|}{\|\nabla_x \ell(x_0)\|} + \frac{\epsilon}{\|\nabla_x \ell(x_0)\|} \Big| \|\nabla_x \ell(x_0 + \delta)\| - \|\nabla_x \ell(x_0)\| \Big|$$

Using the triangle inequality again:

$$\Big| \|\nabla_x \ell(x_0 + \delta)\| - \|\nabla_x \ell(x_0)\| \Big| \leq \|\nabla_x \ell(x_0 + \delta) - \nabla_x \ell(x_0)\| \leq \|\nabla_x^2 \ell\|\|\delta\|$$

This leads to:

$$\|F(\delta) - F(0)\| \leq 2\epsilon \frac{\|\nabla_x^2 \ell(x_0)\|\|\delta\|}{\|\nabla_x \ell(x_0)\|} \tag{31}$$

The Lipschitz constant is:

$$K = 2\epsilon \cdot \frac{\|\nabla_x^2 \ell(x_0)\|}{\|\nabla_x \ell(x_0)\|} \tag{32}$$

Assuming $K < 1$, the fixed point problem converges. $\qquad\square$

## C Appendix: Demonstration $l^p$ Optimal Attack

**Proposition 2.** *For a training sample $x_0$ exhibiting a non-null gradient and a constraint within $B_p(\epsilon)$, the optimal perturbation, denoted as $\delta^\star$, exists and corresponds to the solution of a fixed-point problem: $\delta^\star = F_p(\delta^\star)$. Specifically, we have:*

$$F_p(\delta) = \epsilon \text{sign}(\nabla_x \ell(x_0 + \delta)) \left| \frac{\nabla_x \ell(x_0 + \delta)}{\|\nabla_x \ell(x_0 + \delta)\|_q} \right|^{q-1} \tag{33}$$

*where the $l^q$ norm serves as the dual to $l^p$, i.e., $\frac{1}{p} + \frac{1}{q} = 1$. The absolute value and multiplication operations are element-wise.*

*Proof.* Assuming the same hypotheses as in Appendix A, a maximum exists on the boundary of the $B_p$ ball. We formulate the Lagrangian with the $l^p$ equality constraint:

$$\mathcal{L}_p(\delta, \lambda) = \ell(x_0 + \delta) - \lambda(\|\delta\|_p - \epsilon) \tag{34}$$

The $l^p$ norm is given by:

$$\|\delta\|_p = \left( \sum_i |\delta_i|^p \right)^{\frac{1}{p}} \tag{35}$$

Hence, its derivative is:

$$\frac{\partial}{\partial \delta} \|\delta\|_p = \text{sign}(\delta) \left( \frac{|\delta|}{\|\delta\|_p} \right)^{p-1} \tag{36}$$

The derivatives of the Lagrangian are:

$$\begin{cases} \frac{\partial}{\partial \delta} \mathcal{L}_p = \nabla_x \ell(x_0 + \delta) - \lambda \text{sign}(\delta) \left( \frac{|\delta|}{\|\delta\|_p} \right)^{p-1} = 0 \\ \frac{\partial}{\partial \lambda} \mathcal{L}_p = -(\|\delta\|_p - \epsilon) = 0 \end{cases} \tag{37}$$

Using the dual norm $l^q$ defined with $\frac{1}{p} + \frac{1}{q} = 1 \rightarrow q = \frac{p}{p-1}$, we can characterize $\lambda$ as:

$$\|\nabla_x \ell(x_0 + \delta)\|_q = \frac{|\lambda|}{\|\delta\|_p^{p-1}} (\|\delta\|_p^p)^{\frac{1}{q}} = |\lambda| \tag{38}$$

Substituting into the first derivative of the Lagrangian:

$$\nabla_x \ell(x_0 + \delta) = \pm \|\nabla_x \ell(x_0 + \delta)\|_q \, \text{sign}(\delta) \left( \frac{|\delta|}{\|\delta\|_p} \right)^{p-1} \tag{39}$$

From this, $\delta$ and $\nabla_x \ell(x_0 + \delta)$ have the same sign up to a multiplicative coefficient (i.e., $\pm$):

$$\frac{\nabla_x \ell(x_0 + \delta)}{\|\nabla_x \ell(x_0 + \delta)\|_q} = \pm \left| \frac{\delta}{\|\delta\|_p} \right|^{p-1} \text{sign}(\delta) \tag{40}$$

Extracting $\delta$ and using $\|\delta\|_p = \epsilon$ yields:

$$\delta = \pm \epsilon \text{sign}(\nabla_x \ell(x_0 + \delta)) \times \left( \frac{|\nabla_x \ell(x_0 + \delta)|}{\|\nabla_x \ell(x_0 + \delta)\|_q} \right)^{\frac{1}{p-1}}$$

The solution with the negative sign would yield a locally decreasing loss function, so we take the positive solution. The Lagrange multiplier for maximization is positive:

$$\lambda = \|\nabla_x \ell(x_0 + \delta)\|_q \tag{41}$$

Using $p = \frac{q}{q-1} \rightarrow p - 1 = \frac{1}{q-1}$, we get the final result:

$$\delta = \epsilon \text{sign}(\nabla_x \ell(x_0 + \delta)) \times \left( \frac{|\nabla_x \ell(x_0 + \delta)|}{\|\nabla_x \ell(x_0 + \delta)\|_q} \right)^{q-1}$$

$\square$

## D   Appendix: Lipschitzness of the $l^p$ Fixed-Point Problem

We assume: $\exists m > 0 : \forall \delta \in \partial B_p(\epsilon), |\nabla_\theta \ell(x_0 + \delta)_i| > m$, and proceed to demonstrate Lipschitzness of the function $F_p(\delta)$ verifying the fixed point, defined as:

$$F_p(\delta) = \epsilon \, \text{sign}(\nabla_x \ell(x_0 + \delta)) \left| \frac{\nabla_x \ell(x_0 + \delta)}{\|\nabla_x \ell(x_0 + \delta)\|_q} \right|^{q-1} \tag{42}$$

The sign function can be circumvented by using "one power" of the absolute value of the gradient:

$$F_p(\delta) = \epsilon \frac{\nabla_x \ell(x_0 + \delta)}{\|\nabla_x \ell(x_0 + \delta)\|_q} \times \left( \frac{|\nabla_x \ell(x_0 + \delta)|}{\|\nabla_x \ell(x_0 + \delta)\|_q} \right)^{q-2} \tag{43}$$

The term $q - 2$ is negative, which is permissible since we assumed a lower limit $m$ for gradient values. Our objective is to prove that $F_p(\delta)$ is Lipschitz continuous around $\delta = 0$.

First, let's define:

$$f_q(\delta) = \frac{\nabla_x \ell(x_0 + \delta)}{\|\nabla_x \ell(x_0 + \delta)\|_q} \tag{44}$$

We have:

$$F_p(\delta) = \epsilon f_q(\delta) |f_q(\delta)|^{q-2} \tag{45}$$

Similar to Appendix A, by introducing a cross term we can show that $f$ and $|f|$ are Lipschitz continuous, with a constant $K_f$ such that:

$$|f_q(\delta) - f_q(0)| \leq K_f \|\delta\| \tag{46}$$

The same steps are applied as follows:

$$\||f_q(\delta)| - |f_q(0)|\| \leq \|f_q(\delta) - f_q(0)\| \leq \left\| \frac{\nabla_x \ell(x_0 + \delta)}{\|\nabla_x \ell(x_0 + \delta)\|_q} - \frac{\nabla_x \ell(x_0)}{\|\nabla_x \ell(x_0)\|_q} \right\| \tag{47}$$

By further manipulation and using the triangle inequality:

$$\left| |f_q(\delta)| - |f_q(0)| \right| \le \left\| \frac{\nabla_x \ell(x_0 + \delta)}{\|\nabla_x \ell(x_0 + \delta)\|_q} - \frac{\nabla_x \ell(x_0 + \delta)}{\|\nabla_x \ell(x_0)\|_q} \right\| + \left\| \frac{\nabla_x \ell(x_0)}{\|\nabla_x \ell(x_0)\|_q} - \frac{\nabla_x \ell(x_0 + \delta)}{\|\nabla_x \ell(x_0)\|_q} \right\| \tag{48}$$

This leads to:

$$\left| |f_q(\delta)| - |f_q(0)| \right| \le \left( 1 + \frac{\|\nabla_x \ell(x_0 + \delta)\|}{\|\nabla_x \ell(x_0 + \delta)\|_q} \right) \times \frac{\|\nabla_x^2 \ell(x_0)\|}{\|\nabla_x \ell(x_0)\|_q} \|\delta\| \tag{49}$$

In a finite-dimensional vector space, all norms are equivalent:

$$\exists\, C \ge 0, \ \frac{\|\nabla_x \ell(x_0 + \delta)\|}{\|\nabla_x \ell(x_0 + \delta)\|_q} \le C \tag{50}$$

Next, examining $|x|^{q-2}$ on the interval $[m, +\infty)$ with $q - 2$ negative:

$$\forall (x, y) \in [m, +\infty), \ \left| |x|^{q-2} - |y|^{q-2} \right| \le (2 - q) m^{q-3} |x - y| \tag{51}$$

Using these results for the local Lipschitz continuity of $F_p$:

$$\frac{1}{\epsilon} \|F_p(\delta) - F_p(0)\| = \left\| f_q(\delta) |f_q(\delta)|^{q-2} - f_q(0) |f_q(0)|^{q-2} \right\| \tag{52}$$

Through a series of bounds:

$$\frac{1}{\epsilon} \|F_p(\delta) - F_p(0)\| \le \left\| f_q(\delta) |f_q(\delta)|^{q-2} - f_q(\delta) |f_q(0)|^{q-2} \right\|$$
$$+ \left\| f_q(\delta) |f_q(0)|^{q-2} - f_q(0) |f_q(0)|^{q-2} \right\| \tag{53}$$

Further simplifying:

$$\frac{1}{\epsilon} \|F_p(\delta) - F_p(0)\| \le \frac{\|\nabla_x \ell(x_0 + \delta)\|}{\|\nabla_x \ell(x_0 + \delta)\|_q} \times (2 - q) m^{q-3} |f_q(\delta) - f_q(0)|$$
$$+ \left| \frac{\nabla_x \ell(x_0)}{\|\nabla_x \ell(x_0)\|_q} \right|^{q-2} \times \|f_q(\delta) - f_q(0)\| \tag{54}$$

This yields:

$$\|F_p(\delta) - F_p(0)\| \le K(p, m) \epsilon \times \frac{\|\nabla_x^2 \ell(x_0)\|}{\|\nabla_x \ell(x_0)\|_q} \|\delta\| \tag{55}$$

where:

$$K(p, m) = \left( C(2 - q) m^{q-3} + \left( \frac{m}{\|\nabla_x \ell(x_0)\|_q} \right)^{q-2} \right) (1 + C) \tag{56}$$

# E  Preliminary Validation of Fixed $l^p$ Norms

To understand the fundamental limitations of fixed $p$ values and motivate our adaptive approach, we conducted systematic evaluation of $l^p$-FGSM across different norm values on standard datasets. This preliminary analysis reveals the inherent trade-offs that necessitate adaptive norm selection. All experiments were conducted on a single NVIDIA A100 GPU.

## E.1  Experimental Setup

We evaluate fixed $l^p$-FGSM following the framework of Wong et al. [10] using PGD-50 attacks on CIFAR-10, CIFAR-100 [19], and SVHN [32]. Experiments use PreactResNet18 [33] for SVHN and WideResNet28-10 [20] for CIFAR datasets, with results averaged over five seeds for reliability.

This validation deliberately excludes enhancements like weight decay, dropout, or noise injection to isolate the effects of norm selection and provide a clear baseline for understanding the impact of the $l^p$ norm parameter. All experiments use perturbation radius $\epsilon = 8/255$ for both training and evaluation attacks.

## E.2 Key Findings: The Fixed $p$ Dilemma

Figure 7 presents comprehensive results across all three datasets, revealing several critical insights about the fundamental limitations of fixed norm approaches.

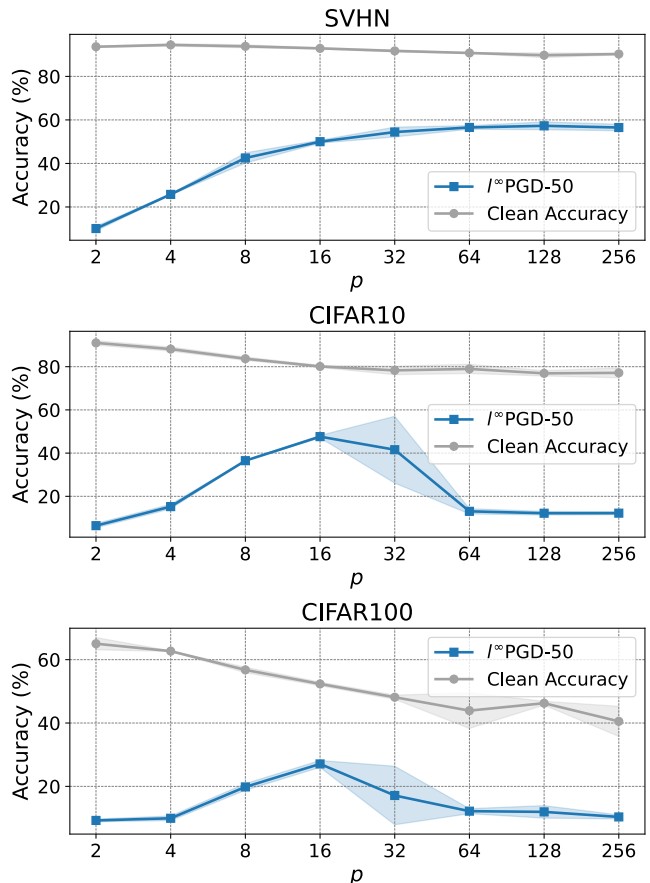

Figure 7: Detailed analysis of clean and adversarial accuracy across CIFAR-10, CIFAR-100, and SVHN datasets with $\epsilon = 8/255$ for different $p$ values. The results demonstrate the fundamental limitations of fixed norm approaches: CIFAR-10 shows optimal performance at intermediate $p \approx 16 - 32$ before CO onset, SVHN exhibits remarkable resilience to CO even at higher $p$ values, while CIFAR-100 displays heightened sensitivity to norm selection with narrow optimal ranges. These dataset-dependent behaviors highlight the critical need for adaptive norm selection.

The results demonstrate striking dataset-dependent optimal ranges that expose the inadequacy of any universal fixed $p$ approach. CIFAR-10 achieves optimal performance at intermediate $p$ values around 16-32, demonstrating a clear sweet spot before catastrophic overfitting occurs. In contrast, SVHN exhibits remarkable resilience to CO even at higher $p$ values, suggesting that simpler datasets can tolerate more aggressive norm constraints for extended periods. CIFAR-100 shows heightened sensitivity to norm selection with narrow optimal ranges, indicating that complex datasets require more conservative and careful norm tuning.

Across all datasets, we observe a universal trade-off pattern that reveals the inherent limitations of fixed approaches. Lower $p$ values ($p \le 4$) provide excellent stability against catastrophic overfitting but at the cost of significantly reduced adversarial robustness. Higher $p$ values ($p \ge 64$) initially improve robustness but eventually lead to catastrophic overfitting, with the onset timing varying dramatically by dataset complexity. Intermediate $p$ values offer the best balance but require careful tuning that is fundamentally dataset-dependent.

The relationship between dataset complexity and optimal norm selection proves particularly striking. Simple datasets like SVHN tolerate aggressive norms for longer periods, while complex datasets

like CIFAR-100 require more conservative norm choices from the outset. This suggests that gradient structure varies significantly across problem domains, with complexity directly influencing the rate at which gradient concentration occurs during training.

## E.3 Fundamental Limitations of Fixed Norm Approaches

These results expose several fundamental limitations of any fixed $p$ approach that render such methods inadequate for general-purpose adversarial training. The lack of generalizability is perhaps most concerning: no single $p$ value works optimally across all datasets, with configurations that succeed for SVHN failing dramatically for CIFAR-100. This dataset dependency makes fixed approaches impractical for real-world deployment where diverse data characteristics are encountered.

The static nature of fixed values conflicts directly with the dynamic nature of adversarial training. Gradient structure evolves throughout training, with early phases potentially benefiting from higher $p$ values while later stages require lower values to prevent catastrophic overfitting. Fixed approaches cannot adapt to these changing conditions, forcing suboptimal compromises throughout the training process.

Even the best fixed $p$ value for each dataset represents a compromise that sacrifices either robustness or stability. The narrow optimal ranges, particularly evident in CIFAR-100, make fixed approaches highly sensitive to hyperparameter selection and prone to overfitting validation performance. This sensitivity creates practical deployment challenges where slight dataset variations can push performance outside optimal ranges.

## E.4 Theoretical Alignment and Motivation for Adaptive Approaches

These empirical observations align perfectly with our gradient concentration hypothesis and provide strong motivation for adaptive norm selection. Complex datasets like CIFAR-100 likely exhibit more concentrated gradients earlier in training, requiring conservative norm choices to prevent early catastrophic overfitting. Simple datasets like SVHN maintain more distributed gradients longer, tolerating aggressive norms without immediate vulnerability. Intermediate complexity datasets like CIFAR-10 require dynamic adaptation as gradient structure evolves throughout training.

The clear dataset dependency and fundamental trade-offs exposed in these experiments provide compelling evidence that fixed norm approaches are inherently limited. An effective solution must automatically adapt to different dataset characteristics without manual tuning, respond to changing gradient structure throughout training, base norm selection on measurable gradient properties that predict catastrophic overfitting onset, and maintain computational efficiency comparable to fixed approaches.

This preliminary analysis establishes the empirical foundation for our theoretical framework and demonstrates why gradient-aware adaptive norm selection is not merely beneficial but necessary for robust fast adversarial training across diverse problem domains. The development of our adaptive $l^p$-FGSM framework detailed in the main paper directly addresses these limitations through principled gradient concentration measurement and automatic norm adaptation.

# F Appendix: Proof of Noise-Induced Alignment

**Lemma 1 (Noise-Induced Alignment).** *For $g \in \mathbb{R}^d$ nonzero and $\eta \sim \mathcal{U}[-M, M]^d$, $\exists \alpha > 0$ such that if $M < \alpha \|g\|_\infty$:*

$$\mathbb{E}\left[\frac{\|g + \eta\|_1}{\|g + \eta\|_2}\right] \geq \frac{\|g\|_1}{\|g\|_2} \tag{57}$$

*Proof.* Let $S_+ = \{i : |g_i| > M\}$ and $S_- = \{i : |g_i| \leq M\}$ partition coordinates.

For $i \in S_+$:

$$\sum_{i \in S_+} |g_i + \eta_i| \geq \sum_{i \in S_+} (|g_i| - M) \tag{58}$$

For $i \in S_-$, direct calculation yields:

$$\mathbb{E}[|g_i + \eta_i|] = \frac{1}{2M} \int_{-M}^{M} |g_i + \eta| \, d\eta = \frac{(g_i + M)^2 + (g_i - M)^2}{4M} = \frac{g_i^2 + M^2}{2M} \quad (59)$$

Thus for the $l^1$ norm:

$$\mathbb{E}[\|g + \eta\|_1] \geq \sum_{i \in S_+} (|g_i| - M) + \sum_{i \in S_-} \frac{g_i^2 + M^2}{2M} \quad (60)$$

For the $l^2$ norm, using $\mathbb{E}[\eta_i^2] = \frac{M^2}{3}$ and independence:

$$\mathbb{E}[\|g + \eta\|_2^2] = \sum_{i=1}^{d} \left( g_i^2 + \frac{M^2}{3} \right) \quad (61)$$

By Jensen's inequality applied to the concave function $f(x) = \sqrt{x}$:

$$\mathbb{E}[\|g + \eta\|_2] = \mathbb{E}\left[ \sqrt{\sum_{i=1}^{d}(g_i + \eta_i)^2} \right]$$

$$\leq \sqrt{\mathbb{E}\left[ \sum_{i=1}^{d}(g_i + \eta_i)^2 \right]}$$

$$= \sqrt{\sum_{i=1}^{d} \left( g_i^2 + \frac{M^2}{3} \right)} \quad (62)$$

Let $\mathcal{E}$ be the event where:

$$\|g + \eta\|_2 \leq \sqrt{\sum_{i=1}^{d} \left( g_i^2 + \frac{M^2}{2} \right)} \quad (63)$$

Then:

$$\mathbb{E}\left[ \frac{\|g + \eta\|_1}{\|g + \eta\|_2} \right] \geq \mathbb{P}(\mathcal{E}) \cdot \frac{\sum_{i \in S_+}(|g_i| - M) + \sum_{i \in S_-} \frac{g_i^2 + M^2}{2M}}{\sqrt{\sum_{i=1}^{d} \left( g_i^2 + \frac{M^2}{2} \right)}} \quad (64)$$

For $M < \alpha \|g\|_\infty$ with $\alpha$ sufficiently small:

- $\mathbb{P}(\mathcal{E})$ approaches 1
- The gain in $S_-$ terms ($\frac{g_i^2 + M^2}{2M} > |g_i|$) exceeds the loss in $S_+$ terms
- The denominator remains close to $\|g\|_2$

Therefore, the ratio exceeds $\frac{\|g\|_1}{\|g\|_2}$. □

## G   Appendix: Proof of Monotonicity of Angular Separation

**Lemma 2 (Monotonicity of Angular Separation).** *For any gradient $\nabla_x \ell$ and $2 \leq p \leq \infty$, the cosine similarity between $l^2$ and $l^p$ perturbations satisfies:*

$$\cos(\theta_{2,p}) \geq \cos(\theta_{2,\infty}) = \sqrt{\frac{\text{PR}_1}{d}} \quad (65)$$

*Proof.* **Step 1: Express** $\cos(\theta_{2,p})$ **in normalized form.**

Let $q = \frac{p}{p-1}$ be the dual exponent of $p$; hence $2 \leq p \leq \infty$ implies $1 \leq q \leq 2$. Recall that:

$$\delta_p = \epsilon \operatorname{sign}(\nabla_x \ell(x_0)) \left| \frac{\nabla_x \ell(x_0)}{\|\nabla_x \ell(x_0)\|_q} \right|^{q-1} \tag{66}$$

$$\delta_\infty = \epsilon \operatorname{sign}(\nabla_x \ell(x_0)) \tag{67}$$

The cosine similarity between the two perturbations is:

$$\cos(\theta_{2,p}) = \frac{\langle \delta_2, \delta_p \rangle}{\|\delta_2\|_2 \|\delta_p\|_2} \tag{68}$$

After computing the inner product and norms, this yields:

$$\cos(\theta_{2,p}) = \frac{\|\nabla_x \ell\|_q^q}{\|\nabla_x \ell\|_2 \|\nabla_x \ell\|_{2(q-1)}^{q-1}} \tag{69}$$

We introduce the normalized vector:

$$g = \frac{\nabla_x \ell}{\|\nabla_x \ell\|_2} \tag{70}$$

Note that $\|g\|_2 = 1$, and each coordinate satisfies $|g_i| \leq 1$. Using $g$:

$$\|\nabla_x \ell\|_q = \|\nabla_x \ell\|_2 \|g\|_q \tag{71}$$

$$\|\nabla_x \ell\|_q^q = \|\nabla_x \ell\|_2^q \|g\|_q^q \tag{72}$$

$$\|\nabla_x \ell\|_{2(q-1)}^{q-1} = \|\nabla_x \ell\|_2^{q-1} \|g\|_{2(q-1)}^{q-1} \tag{73}$$

Substituting these into our expression for $\cos(\theta_{2,p})$:

$$\cos(\theta_{2,p}) = \frac{\|\nabla_x \ell\|_2^q \|g\|_q^q}{\|\nabla_x \ell\|_2 \|\nabla_x \ell\|_2^{q-1} \|g\|_{2(q-1)}^{q-1}} \tag{74}$$

$$= \frac{\|g\|_q^q}{\|g\|_{2(q-1)}^{q-1}} = \frac{\|g\|_q^q}{\sqrt{\|g\|_{2(q-1)}^{2(q-1)}}} \tag{75}$$

**Step 2: Show monotonicity via logarithmic derivative.**

Define:

$$f(q) = \cos(\theta_{2,p}) = \frac{\|g\|_q^q}{\sqrt{\|g\|_{2(q-1)}^{2(q-1)}}} \tag{76}$$

Taking logarithms:

$$\ln f(q) = q \ln \|g\|_q - (q-1) \ln \|g\|_{2(q-1)} \tag{77}$$

For any $l^r$ norm, the derivative with respect to $r$ is:

$$\frac{d}{dr} \ln \|g\|_r = \frac{1}{r} \left( \frac{\sum_i |g_i|^r \ln |g_i|}{\sum_i |g_i|^r} - \ln \|g\|_r \right) \tag{78}$$

Applying this formula to compute $\frac{d \ln f}{dq}$, after simplification (the $\ln \|g\|$ terms cancel):

$$\frac{d \ln f}{dq} = \frac{\sum_i |g_i|^q \ln |g_i|}{\sum_i |g_i|^q} - \frac{\sum_i |g_i|^{2(q-1)} \ln |g_i|}{\sum_i |g_i|^{2(q-1)}} \tag{79}$$

Now we show this derivative is non-negative via convexity. Consider the function:

$$\phi(r) = \ln \|g\|_r^r = \ln \sum_i |g_i|^r \tag{80}$$

Its first derivative is precisely the weighted average that appears above:

$$\phi'(r) = \frac{\sum_i |g_i|^r \ln |g_i|}{\sum_i |g_i|^r} \tag{81}$$

The second derivative, using the quotient rule, is:

$$\phi''(r) = \frac{\sum_i |g_i|^r (\ln |g_i|)^2}{\sum_i |g_i|^r} - \left( \frac{\sum_i |g_i|^r \ln |g_i|}{\sum_i |g_i|^r} \right)^2 \tag{82}$$

$$= \mathrm{Var}_{w^{(r)}}[\ln |g|] \geq 0 \tag{83}$$

where $w_i^{(r)} = |g_i|^r / \sum_j |g_j|^r$. Since variance is always non-negative, $\phi$ is convex, hence $\phi'$ is monotonically increasing.

Observe that $\frac{d \ln f}{dq} = \phi'(q) - \phi'(2(q-1))$.

For $q \in (1, 2]$, we have $2(q-1) \leq q$ (since $2(q-1) = 2q - 2 \leq q$ when $q \leq 2$).
Since $\phi'$ is monotonically increasing and $2(q-1) \leq q$:

$$\phi'(2(q-1)) \leq \phi'(q) \Rightarrow \frac{d \ln f}{dq} = \phi'(q) - \phi'(2(q-1)) \geq 0 \tag{84}$$

This proves that $\cos(\theta_{2,p})$ is monotonically increasing in $q$ (equivalently, decreasing in $p$).

**Step 3: Establish the boundary values using limits.**

At $q = 2$ (corresponding to $p = 2$):

$$\cos(\theta_{2,2}) = \frac{\|g\|_2^2}{\|g\|_2^2} = 1 \tag{85}$$

For the limit as $q \to 1^+$ (corresponding to $p \to \infty$):

$$\lim_{q \to 1^+} \cos(\theta_{2,p}) = \lim_{q \to 1^+} \frac{\|g\|_q^q}{\sqrt{\|g\|_{2(q-1)}^{2(q-1)}}} \tag{86}$$

As $q \to 1^+$: the numerator approaches $\|g\|_1$, and $2(q-1) \to 0^+$. For the denominator, $\lim_{r \to 0^+} \|g\|_r^r = d$ (the number of non-zero components). Therefore:

$$\cos(\theta_{2,\infty}) = \lim_{q \to 1^+} \cos(\theta_{2,p}) = \frac{\|g\|_1}{\sqrt{d}} \tag{87}$$

Since $\cos(\theta_{2,p})$ is monotonically increasing in $q$ (decreasing in $p$), and using:

$$\|g\|_1 = \frac{\|\nabla_x \ell\|_1}{\|\nabla_x \ell\|_2} = \sqrt{\mathrm{PR}_1} \tag{88}$$

we conclude:

$$\frac{\|g\|_1}{\sqrt{d}} = \sqrt{\frac{\mathrm{PR}_1}{d}} = \cos(\theta_{2,\infty}) \leq \cos(\theta_{2,p}) \tag{89}$$

$$\square$$

# H    Appendix: Taylor Expansion of Cosine Similarity

**Proposition 3.** *For $q = 1 + \epsilon$ with small $\epsilon$ and normalized gradient components $\pi_i = \frac{|\nabla_x \ell_i|}{\|\nabla_x \ell\|_1}$, the cosine similarity between $l^2$ and $l^p$ perturbations admits the following first-order expansion:*

$$\cos(\theta_{2,p}) = \sqrt{\frac{\text{PR}_1}{d}} \left(1 + \epsilon(H_m - H)\right) + O(\epsilon^2) \tag{90}$$

*where* $\text{PR}_1 = \left(\frac{\|\nabla_x \ell\|_1}{\|\nabla_x \ell\|_2}\right)^2$ *is the participation ratio, $H$ is the Shannon entropy, and $H_m$ is the logarithmic mean entropy.*

*Proof.* Starting with the cosine similarity for $q = 1 + \epsilon$:

$$\cos(\theta_{2,p}) = \frac{\|\nabla_x \ell\|_q^q}{\|\nabla_x \ell\|_2 \|\nabla_x \ell\|_{2(q-1)}^{q-1}} \tag{91}$$

The numerator expands directly as:

$$\|\nabla_x \ell\|_q^q = \sum_i |\nabla_x \ell_i|^{1+\epsilon} = \|\nabla_x \ell\|_1 \left(1 + \epsilon \sum_i \frac{|\nabla_x \ell_i|}{\|\nabla_x \ell\|_1} \times \log |\nabla_x \ell_i| + O(\epsilon^2)\right) \tag{92}$$

For the denominator term $\|\nabla_x \ell\|_{2\epsilon}^\epsilon$:

$$\|\nabla_x \ell\|_{2\epsilon}^\epsilon = \left(1 + 2\epsilon \sum_i \frac{\log |\nabla_x \ell_i|}{d} + O(\epsilon^2)\right)^{\frac{1}{2}} = 1 + \epsilon \sum_i \frac{\log |\nabla_x \ell_i|}{d} + O(\epsilon^2) \tag{93}$$

Combining terms with normalized gradient components $\pi_i$:

$$\cos(\theta_{2,p}) = \frac{\|\nabla_x \ell\|_1}{\|\nabla_x \ell\|_2 \sqrt{d}} \left(1 + \epsilon \Big(\sum_i \pi_i \log |\nabla_x \ell_i| - \sum_i \frac{\log |\nabla_x \ell_i|}{d}\Big)\right) + O(\epsilon^2) \tag{94}$$

The sums relate to entropy measures through:

$$\sum_i \pi_i \log |\nabla_x \ell_i| = -H + \log \|\nabla_x \ell\|_1 \tag{95}$$

$$\sum_i \frac{\log |\nabla_x \ell_i|}{d} = -H_m + \log \|\nabla_x \ell\|_1 \tag{96}$$

where:

$$H = -\sum_i \pi_i \log(\pi_i) \tag{97}$$

$$H_m = -\log \prod_{i=1}^{d} (\pi_i)^{\frac{1}{d}} \tag{98}$$

Therefore:

$$\cos(\theta_{2,p}) = \sqrt{\frac{\text{PR}_1}{d}} \left(1 + \epsilon(H_m - H)\right) + O(\epsilon^2) \tag{99}$$

The entropy gap $\Delta H = H_m - H$ is always positive by Jensen's inequality. $\qquad\square$

Table 2: CIFAR-10 (WRN-28-8) Clean and AutoAttack Accuracy Evaluation. Results are averaged over multiple seeds. Clean accuracy (top) and AutoAttack accuracy (bottom).

| | CIFAR-10 WRN-28-10 AutoAttack | | | |
|---|---|---|---|---|
| $255 \cdot \epsilon$ | FGSM | RS-FGSM | N-FGSM | $l^p$-FGSM |
| 2 | **90.81**% $\pm$ 0.07 | 90.64% $\pm$ 0.12 | 89.27% $\pm$ 0.21 | 89.02% $\pm$ 0.41 |
| | 74.72% $\pm$ 0.37 | 71.47% $\pm$ 0.44 | 73.14% $\pm$ 0.68 | **76.14**% $\pm$ 0.62 |
| 4 | **87.86**% $\pm$ 0.23 | 86.58% $\pm$ 0.22 | 86.34% $\pm$ 0.36 | 85.71% $\pm$ 0.53 |
| | 61.58% $\pm$ 0.12 | 54.85% $\pm$ 0.16 | 59.81% $\pm$ 0.27 | **62.12**% $\pm$ 0.42 |
| 8 | **84.89**% $\pm$ 1.20 | 80.14% $\pm$ 0.88 | 74.73% $\pm$ 0.46 | 79.81% $\pm$ 0.57 |
| | 0.00% $\pm$ 0.00 | 35.77% $\pm$ 0.24 | 41.65% $\pm$ 0.45 | **42.43**% $\pm$ 0.58 |
| 12 | **80.23**% $\pm$ 0.63 | 61.65% $\pm$ 1.32 | 62.56% $\pm$ 0.73 | 71.12% $\pm$ 0.38 |
| | 0.00% $\pm$ 0.00 | 0.00% $\pm$ 0.00 | 30.17% $\pm$ 1.16 | **32.13**% $\pm$ 0.71 |
| 16 | **74.61**% $\pm$ 0.19 | 69.20% $\pm$ 0.15 | 52.89% $\pm$ 0.27 | 58.43% $\pm$ 0.48 |
| | 0.00% $\pm$ 0.00 | 0.00% $\pm$ 0.00 | 22.50% $\pm$ 0.89 | **25.89**% $\pm$ 0.59 |

# I   Appendix: AutoAttack Results

To ensure a comprehensive assessment, we have also included robust accuracy results evaluated with AutoAttack (AA) [34]. We present the clean (top) and robust (bottom) accuracies (3 seeds) for CIFAR-10 using WRN-28-8, evaluated with AA. The pattern observed is consistent with the results from PGD-50, showing a common trend.

The comparison encompasses standard FGSM [5], RS-FGSM [10], N-FGSM with (k=2) [13], and our proposed adaptive $l^p$-FGSM ($\beta = 0.01$). The experiments reveal a characteristic pattern of Catastrophic Overfitting (CO) across various perturbation magnitudes ($\epsilon$) for FGSM and RS-FGSM. During CO, models maintain high clean accuracy while their robust accuracy against adversarial attacks deteriorates to near zero.

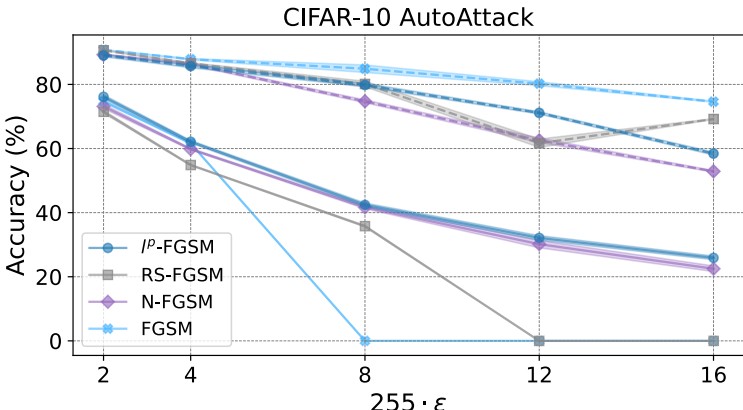

Figure 8: Comparative evaluation using AutoAttack on CIFAR-10 with WideResNet-28-10 across different perturbation magnitudes. Results demonstrate consistent robustness assessment between PGD-50 and AutoAttack [34], validating the reliability of our evaluation methodology.

The strong agreement between PGD-50 and AutoAttack results strengthens our evaluation methodology, as AutoAttack combines multiple complementary attack strategies [34, 11]. This comprehensive assessment validates our findings regarding the effectiveness of norm selection in preventing CO.

## J    Appendix: Long-Term Training Evaluation

To rigorously assess the durability and stability of the $l^p$-FGSM method under prolonged training conditions, we conducted an extended training experiment spanning 200 epochs. This experiment utilized the CIFAR-10 dataset with adversarial perturbation norms set at $\epsilon = 8/255$ and $\epsilon = 16/255$, using ADAM optimizer with a learning rate of 0.001.

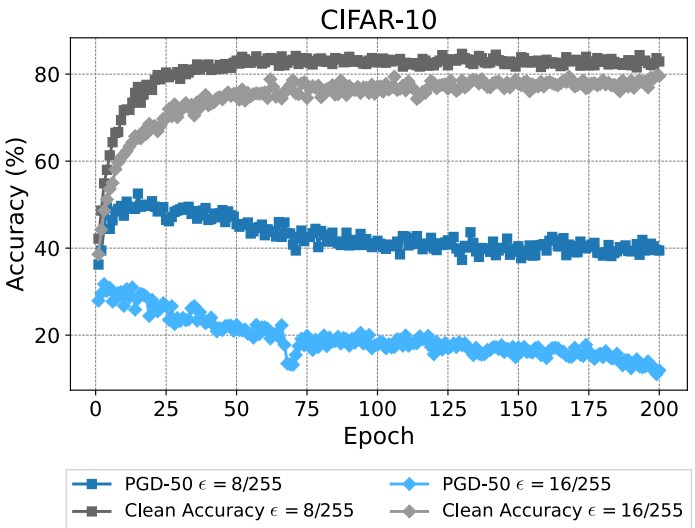

Figure 9: Extended training performance of $l^p$-FGSM on CIFAR-10. While Catastrophic Overfitting (CO) was not observed, the experiment highlights the occurrence of robust overfitting over a prolonged training period.

The results of this long-term training provide insightful observations. Crucially, no instances of Catastrophic Overfitting (CO) were detected throughout the training process, underscoring the robustness of the $l^p$-FGSM approach. However, a slight decrease in robustness, i.e., robust overfitting, occurs. This occurrence warrants early stopping and cyclical learning rates to offset this phenomenon.

# K   Appendix: $l^p$-FGSM Results Tables

Table 3: Comparative Analysis of Fast Adversarial Training Methods on SVHN Dataset

| | | SVHN PreAct-18 PGD-50-10 | | | |
|---|---|---|---|---|---|
| $\epsilon \cdot 255$ | $l^p$-FGSM | RS-FGSM | N-FGSM | GradAlign | ZeroGrad |
| 2 | 94.20% ±0.52 | **96.16%** ±0.13 | 96.04% ±0.24 | 96.01% ±0.25 | 96.08% ±0.22 |
| | **86.22%** ±0.22 | 86.17% ±0.17 | 86.46% ±0.12 | 86.44% ±0.15 | 86.47% ±0.17 |
| 4 | 94.16% ±0.64 | **95.07%** ±0.08 | 94.56% ±0.18 | 94.57% ±0.24 | 94.83% ±0.19 |
| | **77.86%** ±0.75 | 71.25% ±0.43 | 72.54% ±0.21 | 72.18% ±0.22 | 71.64% ±0.24 |
| 6 | 92.26% ±0.65 | **95.16%** ±0.48 | 92.27% ±0.36 | 92.55% ±0.26 | 93.52% ±0.24 |
| | **64.12%** ±1.27 | 0.00% ±0.00 | 58.44% ±0.18 | 57.36% ±0.27 | 51.77% ±0.58 |
| 8 | 91.06% ±0.69 | **94.48%** ±0.18 | 89.59% ±0.48 | 90.16% ±0.36 | 92.43% ±1.33 |
| | **56.72%** ±0.74 | 0.00% ±0.00 | 45.64% ±0.21 | 43.88% ±0.16 | 35.96% ±2.78 |
| 10 | 90.76% ±1.21 | **93.82%** ±0.28 | 86.78% ±0.88 | 87.26% ±0.73 | 90.36% ±0.33 |
| | **45.46%** ±1.04 | 0.00% ±0.00 | 33.98% ±0.48 | 32.88% ±0.36 | 21.36% ±0.37 |
| 12 | 90.02% ±0.38 | **92.72%** ±0.56 | 81.49% ±1.66 | 84.12% ±0.44 | 88.11% ±0.47 |
| | **36.88%** ±1.09 | 0.00% ±0.00 | 26.17% ±0.88 | 23.64% ±0.42 | 14.16% ±0.38 |

Table 4: Comparative Analysis of Fast Adversarial Training Methods on CIFAR-10 Dataset

| | | CIFAR-10 WRN-28-10 PGD-50-10 | | | |
|---|---|---|---|---|---|
| $\epsilon \cdot 255$ | $l^p$-FGSM | RS-FGSM | N-FGSM | GradAlign | ZeroGrad |
| 2 | 91.12% ±0.52 | **92.86%** ±0.14 | 92.49% ±0.14 | 92.54% ±0.13 | 92.62% ±0.16 |
| | 80.84% ±0.25 | **80.91%** ±0.14 | 81.42% ±0.34 | 81.32% ±0.43 | 81.41% ±0.32 |
| 4 | 88.07% ±0.34 | **90.74%** ±0.23 | 89.64% ±0.23 | 89.93% ±0.34 | 90.21% ±0.22 |
| | **69.62%** ±0.84 | 68.24% ±0.19 | 69.10% ±0.27 | 69.80% ±0.48 | 69.21% ±0.21 |
| 6 | 83.23% ±0.46 | **88.25%** ±0.22 | 85.74% ±0.32 | 86.94% ±0.16 | 86.11% ±0.45 |
| | **59.24%** ±0.51 | 57.24% ±0.19 | 58.26% ±0.18 | 59.14% ±0.16 | 58.44% ±0.19 |
| 8 | 81.67% ±0.61 | 83.61% ±1.77 | 81.64% ±0.35 | 82.16% ±0.21 | **84.16%** ±0.21 |
| | **51.31%** ±0.59 | 0.00% ±0.00 | 49.51% ±0.27 | 50.12% ±0.17 | 48.32% ±0.21 |
| 10 | 76.61% ±0.58 | **82.17%** ±1.48 | 76.94% ±0.12 | 79.42% ±0.28 | 81.29% ±0.73 |
| | **45.87%** ±0.68 | 0.00% ±0.00 | 42.39% ±0.39 | 41.42% ±0.52 | 36.18% ±0.19 |
| 12 | 72.84% ±0.54 | 78.64% ±0.74 | 72.18% ±0.17 | 73.72% ±0.82 | **79.33%** ±0.92 |
| | **41.09%** ±1.24 | 0.00% ±0.00 | 36.82% ±0.27 | 35.16% ±0.77 | 28.26% ±1.81 |
| 14 | 66.58% ±0.63 | 73.27% ±2.84 | 67.86% ±0.46 | 66.41% ±0.52 | **78.18%** ±0.66 |
| | **38.65%** ±0.81 | 0.00% ±0.00 | 31.68% ±0.68 | 30.85% ±0.34 | 18.56% ±0.35 |
| 16 | 63.84% ±0.76 | 68.68% ±2.43 | 56.75% ±0.44 | 57.88% ±0.74 | **75.43%** ±0.89 |
| | **37.16%** ±1.22 | 0.00% ±0.00 | 25.11% ±0.43 | 26.24% ±0.43 | 14.66% ±0.22 |

Table 5: Comparative Analysis of Fast Adversarial Training Methods on CIFAR-100 Dataset

| | | CIFAR-100 WRN-28-10 PGD-50-10 | | | |
|---|---|---|---|---|---|
| $\epsilon \cdot 255$ | $l^p$-FGSM | RS-FGSM | N-FGSM | GradAlign | ZeroGrad |
| 2 | 66.42% ±0.15 | **72.62%** ±0.24 | 71.52% ±0.14 | 71.61% ±0.23 | 71.64% ±0.22 |
| | **55.29%** ±0.64 | 51.62% ±0.56 | 52.24% ±0.35 | 51.51% ±0.48 | 52.63% ±0.64 |
| 4 | 61.32% ±0.34 | **68.27%** ±0.21 | 66.51% ±0.48 | 67.09% ±0.19 | 67.21% ±0.18 |
| | **45.73%** ±0.46 | 39.56% ±0.14 | 39.96% ±0.31 | 39.81% ±0.48 | 39.61% ±0.32 |
| 6 | 58.79% ±0.45 | **65.62%** ±0.66 | 61.42% ±0.63 | 62.86% ±0.10 | 63.65% ±0.12 |
| | **38.33%** ±0.54 | 26.61% ±2.79 | 30.99% ±0.27 | 32.11% ±0.24 | 30.28% ±0.51 |
| 8 | 53.46% ±0.58 | 54.28% ±5.92 | 56.42% ±0.65 | 58.55% ±0.41 | **60.78%** ±0.24 |
| | **32.41%** ±1.18 | 0.00% ±0.00 | 26.71% ±0.68 | 26.97% ±0.61 | 23.72% ±0.16 |
| 10 | 50.23% ±0.42 | 46.18% ±4.88 | 51.51% ±0.61 | 53.85% ±0.73 | **61.11%** ±0.39 |
| | **27.12%** ±0.76 | 0.00% ±0.00 | 23.11% ±0.49 | 22.64% ±0.61 | 15.15% ±0.45 |
| 12 | 47.23% ±0.28 | 35.86% ±0.27 | 46.42% ±0.56 | 46.94% ±0.86 | **58.36%** ±0.15 |
| | **24.74%** ±0.67 | 0.00% ±0.00 | 19.32% ±0.51 | 19.94% ±0.65 | 11.12% ±0.66 |
| 14 | 43.18% ±0.25 | 24.42% ±1.38 | 42.14% ±0.36 | 42.63% ±0.50 | **56.24%** ±0.16 |
| | **22.32%** ±1.13 | 0.00% ±0.00 | 16.62% ±0.44 | 16.96% ±0.14 | 8.81% ±0.34 |
| 16 | 40.56% ±1.64 | 21.47% ±5.21 | 38.37% ±0.48 | 36.17% ±0.45 | **56.42%** ±0.29 |
| | **18.41%** ±1.42 | 0.00% ±0.00 | 14.29% ±0.38 | 14.23% ±0.26 | 4.92% ±0.38 |

# L    Appendix: Effects of $\varepsilon$-Softening and Noise Injection

We investigate two key components of our $l^p$-FGSM framework: the $\varepsilon$-softening term from Algorithm 1 and the integration of random noise.

The $\varepsilon$-softening term, introduced to maintain Lipschitz continuity in our fixed-point formulation, helps numerical stability by avoiding zero division. Furthermore, there is a contrast with ZeroGrad [12] that nullifies small gradient components, while our softening ensures gradients maintain minimal non-zero values.

The theoretical motivation behind $\varepsilon$-softening stems from the observation that the fixed-point mapping's contractiveness is particularly sensitive near zero-gradient regions. By introducing a small, non-zero floor to gradient magnitudes, we maintain the desirable theoretical properties of our fixed-point formulation while improving numerical stability [11, 35].

For noise integration, following [10], we can employ a dual-purpose strategy where noise can either serve as input augmentation or initialization for perturbation crafting:

$$\begin{cases} x_0 \leftarrow x_0 + \eta, \ \eta \sim \mathcal{U}[-\epsilon, \epsilon] \\ \delta_0 \leftarrow \Pi_{\partial B_p(\epsilon)}(\eta) \end{cases} \tag{100}$$

These two noise placement approaches can be used independently. The random initialization at boundary $\partial B_p(\epsilon)$ particularly helps when gradient information is near zero. Our implementation differs from previous approaches in two key aspects: first, we project the noise onto the $l^p$ ball boundary rather than using uniform sampling, and second, we reuse the same noise vector for both input augmentation and initialization, reducing computational overhead [36].

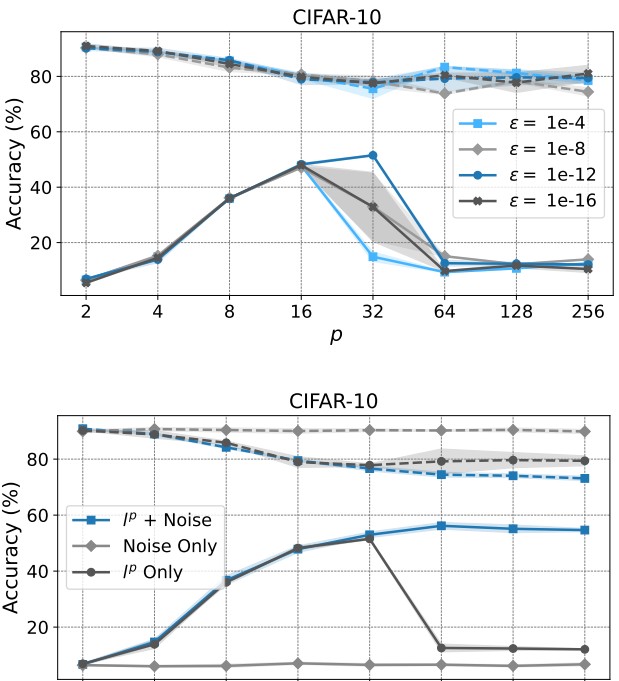

Figure 10: Analysis of $\varepsilon$-softening and noise effects on CIFAR-10 using WideResNet-28-10 against PGD-50 ($\epsilon = 8/255$). Top: Effect of $\varepsilon$-softening on clean (dashed) and adversarial (solid) accuracy for various $p$ values. Optimal $\varepsilon$ enhances stability against CO. Bottom: Synergistic effects of noise injection showing improved robustness against CO and enhanced overall accuracy. The results demonstrate that both components contribute significantly to preventing catastrophic overfitting while maintaining competitive performance.

Even though the main paper does not use any noise, the synergistic relationship between $\varepsilon$-softening and noise injection becomes apparent in their complementary effects on training stability. While $\varepsilon$-softening provides consistent gradient behavior, noise injection helps explore the loss landscape more effectively [34]. This combination proves particularly effective in preventing the gradient collapse often associated with CO [11].

Our extensive experiments on CIFAR-10 with WideResNet-28-10 (Figure 10) demonstrate that both components contribute meaningfully to the algorithm's performance. The $\varepsilon$-softening exhibits an optimal range where it enhances stability without compromising accuracy, while noise injection provides complementary benefits in preventing CO and improving overall robustness.

Notably, we observe that the combination of these techniques allows for more aggressive training schedules than previously possible [10, 37], achieving faster convergence while maintaining robustness. These findings suggest promising directions for future research in stabilizing adversarial training in conjunction with our adaptive $l^p$-FGSM.

# M   Appendix: Entropy Gap and $\mathrm{PR}_1$ for $l^\infty$ vs $l^p$

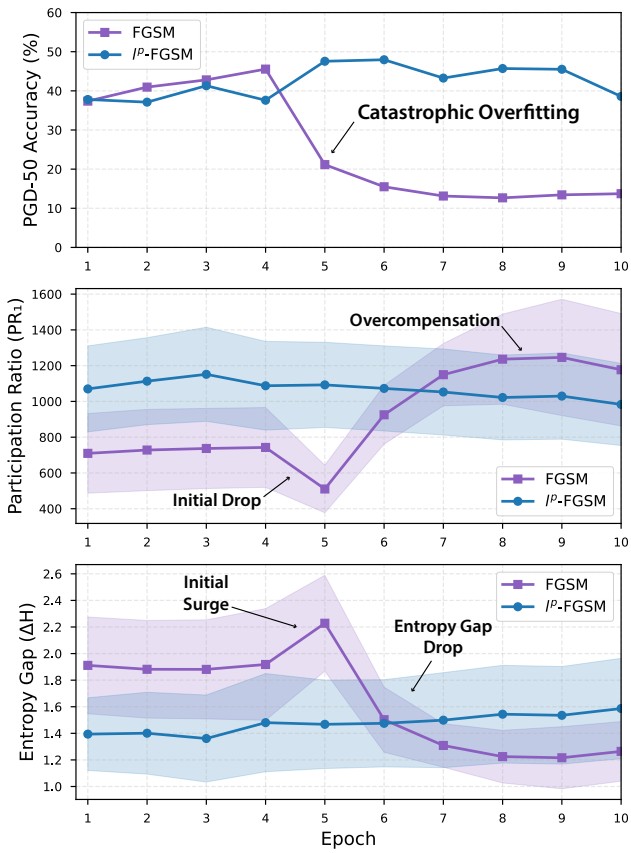

Figure 11: Evolution of Participation Ratios ($\mathrm{PR}_1$) and entropy gap during training with and without $l^p$-FGSM. Sharp patterns in these metrics align with the onset of Catastrophic Overfitting (CO), highlighting the link between gradient concentration and adversarial vulnerability. Same experimental setting as Figure 4.

Our preliminary analysis suggests that gradient concentration metrics (Participation Ratio and entropy gap) exhibit notable changes that appear to coincide with the onset of Catastrophic Overfitting. As shown in Figure 11, these metrics display an interesting pattern that warrants further investigation: a moderate increase, followed by a drop, and then what appears to be a compensatory response. While more extensive experimentation is needed to fully validate these observations, the pattern is consistent across multiple experimental runs.

The adaptation of Participation Ratio (PR) from quantum mechanics [21, 22] to the adversarial training context as $PR_1$ represents a novel approach to quantifying gradient behavior. In quantum systems, PR measures the effective number of states occupied by an electron; similarly, our $PR_1$ aims to capture the effective dimensionality of gradient information. The entropy gap metric offers a complementary perspective, potentially providing insights into how information is distributed across gradient dimensions.

The observed pattern—initial increase, decline, and subsequent adjustment—may offer preliminary insights into the dynamics preceding CO. This behavior could potentially reflect the model's changing gradient geometry as it negotiates the complex loss landscape during adversarial training. The initial increase in both $PR_1$ and entropy gap might suggest a temporary distribution of gradient information before concentration occurs.

By leveraging these metrics during training, our adaptive norm selection approach aims to detect potential instabilities and adjust accordingly. While our current results are promising, we acknowledge that the full relationship between these information-theoretic measures and adversarial robustness requires deeper exploration.

These initial findings provide support for our theoretical framework connecting gradient geometry to norm selection, suggesting that the $l^p$-FGSM approach may effectively mitigate CO without requiring additional techniques like gradient alignment or noise injection. Future work could explore these connections more thoroughly, potentially yielding broader insights into neural network behavior under adversarial constraints.

