# OpenReview forum: "Adaptive Norm Selection Prevents Catastrophic Overfitting in Fast Adversarial Training"
_NeurIPS.cc/2025/Workshop/Reliable_ML — NeurIPS 2025 - Reliable ML Workshop_

### Official Review · Reviewer_4Aax · 2025-09-18
**Rigorous mathematical framework and experimental evaluation**

**Rating:** 7
**Confidence:** 1

**Review:**

### Summary:

The paper presents a framework as a solution to catastrophic overfitting in fast adversarial training. The framework is based on adaptive norm selection under a local convexity hypothesis. The authors present a mathematical foundation and a thorough experimental evaluation of the proposed $l^p$-FGSM algorithm.

### Strengths:

S1. The local convexity assumption enables a fixed point formulation for $l^p$ norms, providing a mathematical foundation for preventing catastrophic overfitting.

S2. The paper has a sound mathematical framework and is quite elegant.

### Weaknesses:

W1. The paper states that the method offers computational efficiency over competitors, the paper does not provide a comparison beyond accuracy.

W2. The parametrization details on the experimental comparison, specifically how the norm parameter $p$ was chosen, could be explained more clearly. How does this compare to the baseline methods configurations?

### Suggestions:

- Rewrite the 3.3 section to have a smoother logical flow. Especially noise-induced alignment and monotonic angular relationships. These insights could follow more naturally, probably after the algorithm.

- It is not clear to me if this approach generalizes well to multiple or mixed-norm attacks. This might be an interesting thing to study.

- Also, in the noise-induced alignment, what happens under different noise models?

---

### Official Review · Reviewer_B5dn · 2025-09-19
**Review of Submission 30**

**Rating:** 9
**Confidence:** 4

**Review:**

**Summary**

The paper presents a solution to catastrophic overfitting in fast adversarial training that doesn’t use noise injection or expensive regularization. Instead, the dynamically adjusts training norms based on gradient concentration. This is inspired by their preliminary findings on how catastrophic overfitting is norm-dependent. Using adaptive l-p norm selection, they are able to 1) surpass robustness benchmarks of leading fast methods and 2) achieve superior adversarial robustness across all perturbation levels while maintaining competitive clean accuracy on ImageNet.

**Strengths**

The development of the method is clearly justified, problem well motivated. Adequate comparisons with leading methods and ablations are done to show highlight their positive result. Narrative flow in introduction is well done.

**Weaknesses**

This paper could benefit from clearer definitions, more precise language, and elaborating some of the comparisons. Some points that could use more clarification:
- Existing methods use clipping, noise injection, and regularization, whereas your methods uses norm. Why is your method better fundamentally? How is it more “principled”? Or is it better because it’s more computationally efficient or something else?- A more formal definition of catastrophic overfitting would help, especially in understanding figure 1 where it’s unclear how you decided where CO Onset happens and how to detect catastrophic overfitting by some definition.
- Figure 1 legend and setup could be clearer. Explain what the X vs. Y in the legend mean. The Y specifies the attack (at test time?) but doesn’t specify the attack at training time.
Explain how to interpret/detect CO from the figure 2.

**Suggestions**
See point 3.

**Ethics (if applicable)**
N/a.